# A neurotrophin functioning with a Toll regulates structural plasticity in a dopaminergic circuit

Jun Sun[1], Francisca Rojo-Cortes[1], Suzana Ulian-Benitez[1], Manuel G Forero[2], Guiyi Li[1], Deepanshu ND Singh[1†], Xiaocui Wang[1], Sebastian Cachero[3], Marta Moreira[1], Dean Kavanagh[4], Gregory SXE Jefferis[3], Vincent Croset[5], Alicia Hidalgo[1]*

[1]Birmingham Centre for Neurogenetics, School of Biosciences, University of Birmingham, Birmingham, United Kingdom; [2]Semillero Lún, Grupo D+Tec, Universidad de Ibagué, Ibagué, Colombia; [3]MRC LMB, Cambridge, United Kingdom; [4]Institute of Biomedical Research, University of Birmingham, Birmingham, United Kingdom; [5]Department of Biosciences, Durham University, Durham, United Kingdom

*For correspondence:
a.hidalgo@bham.ac.uk

Present address: [†]University of Manchester, Manchester, United Kingdom

Competing interest: The authors declare that no competing interests exist.

## eLife Assessment

This **important** study identifies neurotrophin signalling as a molecular mechanism underlying previous findings of structural plasticity in central dopaminergic neurons of the adult fly brain. The authors present **solid** evidence for neurotrophin signalling in shaping the structure and synapses of certain dopaminergic circuits. The work suggests an intriguing potential link between neurotrophin signaling and experience-induced structural plasticity, but further research will be necessary to establish this connection definitively.

**Abstract** Experience shapes the brain as neural circuits can be modified by neural stimulation or the lack of it. The molecular mechanisms underlying structural circuit plasticity and how plasticity modifies behaviour are poorly understood. Subjective experience requires dopamine, a neuromodulator that assigns a value to stimuli, and it also controls behaviour, including locomotion, learning, and memory. In *Drosophila*, Toll receptors are ideally placed to translate experience into structural brain change. *Toll-6* is expressed in dopaminergic neurons (DANs), raising the intriguing possibility that Toll-6 could regulate structural plasticity in dopaminergic circuits. *Drosophila* neurotrophin-2 (DNT-2) is the ligand for Toll-6 and Kek-6, but whether it is required for circuit structural plasticity was unknown. Here, we show that *DNT-2*-expressing neurons connect with DANs, and they modulate each other. Loss of function for *DNT-2* or its receptors *Toll-6* and kinase-less Trk-like *kek-6* caused DAN and synapse loss, impaired dendrite growth and connectivity, decreased synaptic sites, and caused locomotion deficits. In contrast, over-expressed *DNT-2* increased DAN cell number, dendrite complexity, and promoted synaptogenesis. Neuronal activity modified DNT-2, increased synaptogenesis in DNT-2-positive neurons and DANs, and over-expression of DNT-2 did too. Altering the levels of DNT-2 or Toll-6 also modified dopamine-dependent behaviours, including locomotion and long-term memory. To conclude, a feedback loop involving dopamine and DNT-2 highlighted the circuits engaged, and DNT-2 with Toll-6 and Kek-6 induced structural plasticity in this circuit modifying brain function and behaviour.

## Introduction

The brain can change throughout life as new cells are formed or eliminated, axonal and dendritic arbours can grow or shrink, and synapses can form or be eliminated (*Wiesel, 1982*; *Feldman and Brecht, 2005*; *Holtmaat and Svoboda, 2009*; *Gage, 2019*). Such changes can be driven by experience, that is, neuronal activity or the lack of it (*Wiesel, 1982*; *Maguire et al., 2000*; *Cotman and Berchtold, 2002*; *Feldman and Brecht, 2005*; *Sur and Rubenstein, 2005*; *Holtmaat and Svoboda, 2009*; *Woollett and Maguire, 2011*; *Chen and Brumberg, 2021*; *Bharmauria et al., 2022*). Structural changes result in remodelling of connectivity patterns, and these bring about modifications of behaviour. These can be adaptive, dysfunctional, or simply the consequence of opportunistic connections between neurons (*Kuner and Flor, 2016*; *Leemhuis et al., 2019*; *Yang et al., 2020*). It is critical to understand how structural modifications to cells influence brain function. This requires linking with cellular resolution molecular mechanisms, neural circuits, and resulting behaviours.

In the mammalian brain, the neurotrophins (NTs: BDNF, NGF, NT3, NT4) are growth factors underlying structural brain plasticity (*Poo, 2001*; *Lu et al., 2005*; *Park and Poo, 2013*). They promote neuronal survival, connectivity, neurite growth, synaptogenesis, synaptic plasticity, and long-term potentiation (LTP) via their Trk and p75[NTR] receptors (*Poo, 2001*; *Lu et al., 2005*; *Park and Poo, 2013*). In fact, all anti-depressants function by stimulating production of BDNF and signalling via its receptor TrkB, leading to increased brain plasticity (*Casarotto et al., 2021*; *Castrén and Monteggia, 2021*). Importantly, NTs have dual functions and can also induce neuronal apoptosis, neurite loss, synapse retraction, and long-term depression (LTD) via p75[NTR] and Sortilin (*Lu et al., 2005*). Remarkably, these latter functions are shared with neuroinflammation, which in mammals involves Toll-like receptors (TLRs) (*Squillace and Salvemini, 2022*). TLRs and Tolls have universal functions in innate immunity across the animals (*Gay and Gangloff, 2007*), and consistently with this, TLRs in the CNS are mostly studied in microglia. However, mammalian *TLRs* are expressed in all CNS cell types, where they can promote not only neuroinflammation, but also neurogenesis, neurite growth, and synaptogenesis and regulate memory – independently of pathogens, cellular damage, or disease (*Ma et al., 2006*; *Rolls et al., 2007*; *Okun et al., 2010*; *Okun et al., 2011*; *Patel et al., 2016*; *Chen et al., 2019*). Whether TLRs have functions in structural brain plasticity and behaviour remains little explored, and whether they can function together with NTs in the mammalian brain is unknown.

Progress linking cellular and molecular events to circuit and behavioural modification has been rather daunting and limited using mammals (*Wang et al., 2022*). The *Drosophila* adult brain is plastic and can be modified by experience and neuronal activity (*Technau, 1984*; *Barth and Heisenberg, 1997*; *Barth et al., 1997*; *Sachse et al., 2007*; *Kremer et al., 2010*; *Sugie et al., 2015*; *Linneweber et al., 2020*; *Baltruschat et al., 2021*; *Çoban et al., 2024*). Different living conditions, stimulation with odorants or light, circadian rhythms, nutrition, long-term memory, and experimentally activating or silencing neurons modify brain volume, alter circuit and neuronal shape, and remodel synapses, revealing experience-dependent structural plasticity (*Heisenberg et al., 1995*; *Barth and Heisenberg, 1997*; *Barth et al., 1997*; *Devaud et al., 2001*; *Górska-Andrzejak et al., 2005*; *Sachse et al., 2007*; *Fernández et al., 2008*; *Kremer et al., 2010*; *Bushey et al., 2011*; *Sugie et al., 2015*; *Duhart et al., 2020*; *Baltruschat et al., 2021*; *Vaughen et al., 2022*; *Çoban et al., 2024*). Furthermore, the *Drosophila* brain is also susceptible to neurodegeneration (*Bolus et al., 2020*). However, the molecular and circuit mechanisms underlying structural brain plasticity are mostly unknown in *Drosophila*.

*Toll* receptors are expressed across the *Drosophila* brain, in distinct but overlapping patterns that mark the anatomical brain domains (*Li et al., 2020*). Tolls share a common signalling pathway downstream that can drive at least four distinct cellular outcomes – cell death, survival, quiescence, and proliferation – depending on context (*McIlroy et al., 2013*; *Foldi et al., 2017*; *Anthoney et al., 2018*; *Li et al., 2020*). They are also required for connectivity and structural synaptic plasticity, and they can also induce cellular events independently of signalling (*McIlroy et al., 2013*; *Ward et al., 2015*; *McLaughlin et al., 2016*; *Foldi et al., 2017*; *Ulian-Benitez et al., 2017*; *Li et al., 2020*). These nervous system functions occur in the absence of tissue damage or infection. This is consistent with the fact that – as well as universal functions in innate immunity – Tolls also have multiple non-immune functions also outside the CNS, including the original discovery of Toll in dorsoventral patterning, cell intercalation, cell competition, and others (*Meyer et al., 2014*; *Paré et al., 2014*; *Anthoney et al., 2018*; *Tamada et al., 2021*). The Toll distribution patterns in the adult brain and their ability to switch

between distinct cellular outcomes mean they are ideally placed to translate experience into structural brain change (*Li et al., 2020*).

We had previously observed that in the adult brain *Toll-6* is expressed in dopaminergic neurons (DANs) (*McIlroy et al., 2013*). Dopamine is a key neuromodulator that regulates wakefulness and motivation, experience valence, such as reward, and is essential for locomotion, learning, and memory (*Riemensperger et al., 2011*; *Waddell, 2013*; *Adel and Griffith, 2021*). In *Drosophila*, DANs form an associative neural circuit together with mushroom body Kenyon cells (KCs), dorsal anterior lateral neurons (DAL), and mushroom body output neurons (MBONs) (*Chen et al., 2012*; *Aso et al., 2014a*; *Boto et al., 2020*; *Adel and Griffith, 2021*). KCs receive input from projection neurons of the sensory systems and then project through the mushroom body lobes where they are intersected by DANs to regulate MBONs to drive behaviour (*Heisenberg, 2003*; *Aso et al., 2014b*; *Boto et al., 2020*). This associative circuit is required for learning, long-term memory, and goal-oriented behaviour (*Chen et al., 2012*; *Guven-Ozkan and Davis, 2014*; *Adel and Griffith, 2021*). During experience, involving sensory stimulation from the external world and from own actions, dopamine assigns a value to otherwise neutral stimuli, labelling the neural circuits engaged (*Boto et al., 2020*). Thus, this raises the possibility that a link of Toll-6 to dopamine could enable translating experience into circuit modification to modulate behaviour.

In *Drosophila*, Toll receptors can function both independently of ligand-binding and binding Spätzle (Spz) protein family ligands, also known as *Drosophila* neurotrophins (DNTs), which are sequence, structural, and functional homologues of the mammalian NTs (*DeLotto and DeLotto, 1998*; *Weber et al., 2003*; *Hoffmann et al., 2008a*; *Hoffmann et al., 2008b*; *Zhu et al., 2008*; *Lewis et al., 2013*; *McIlroy et al., 2013*; *Foldi et al., 2017*). Like mammalian NTs, DNTs also promote cell survival, connectivity, synaptogenesis, and structural synaptic plasticity, and can also promote cell death, depending on context (*Zhu et al., 2008*; *McIlroy et al., 2013*; *Sutcliffe et al., 2013*; *Foldi et al., 2017*; *Ulian-Benitez et al., 2017*). As well as Tolls, DNTs are also ligands for Kekkon (Kek) receptors, kinase-less homologues of the mammalian NT Trk receptors, and are required for structural synaptic plasticity (*Ulian-Benitez et al., 2017*). Importantly, the targets regulated by Tolls and Keks – ERK, NFκB, PI3K, JNK, CaMKII – are shared with those of mammalian NT receptors Trk and p75$^{NTR}$, and have key roles in structural and functional plasticity across the animals (*Park and Poo, 2013*; *Foldi et al., 2017*; *Ulian-Benitez et al., 2017*; *Yang et al., 2020*; *Tamada et al., 2021*).

Here, we focus on *Drosophila* neurotrophin-2 (DNT-2), proved to be the ligand of Toll-6 and Kek-6, with in vitro, cell culture, and in vivo evidence (*McIlroy et al., 2013*; *Foldi et al., 2017*; *Ulian-Benitez et al., 2017*). Here, we asked how DNT-2, Toll-6, and Kek-6 are functionally related to dopamine, whether they and neuronal activity – as a proxy for experience – can modify neural circuits, and how structural circuit plasticity modifies dopamine-dependent behaviours.

## Results

### DNT-2A, Toll-6, and Kek-6 neurons are integrated in a dopaminergic circuit

To allow morphological and functional analyses of *DNT-2*-expressing neurons, we generated a *DNT-2Gal4* line using CRISPR/Cas9 and drove expression of the membrane-tethered-GFP *FlyBow1.1* reporter. We identified at least 12 DNT-2+ neurons and focused on four anterior DNT-2A neurons per hemi-brain (*Figure 1A and B*). Using the post-synaptic marker Denmark, DNT-2A dendrites were found at the prow (PRW) and flange (FLA) region (*Figure 1C and C'*), whereas axonal terminals visualised with the pre-synaptic marker synapse defective 1 (Dsyd1-GFP) resided at the superior medial protocerebrum (SMP) (*Figure 1C and C''*). We additionally found post-synaptic signal at the SMP and pre-synaptic signal at the FLA/PRW (*Figure 1C, C' and C''*), suggesting bidirectional communication at both sites. Using Multi-Colour Flip-Out (MCFO) to label individual cells stochastically (*Nern et al., 2015*; *Costa et al., 2016*), single-neuron clones revealed variability in the DNT-2A projections across individual flies (*Figure 1D*), consistently with developmental and activity-dependent structural plasticity in *Drosophila* (*Heisenberg et al., 1995*; *Kremer et al., 2010*; *Sugie et al., 2015*; *Mayseless et al., 2018*; *Li et al., 2020*; *Linneweber et al., 2020*; *Baltruschat et al., 2021*). We found that DNT-2A neurons are glutamatergic as they express the vesicular glutamate transporter vGlut (*Figure 1E*, *Figure 1—figure supplement 1A*) and lack markers for other neurotransmitter types

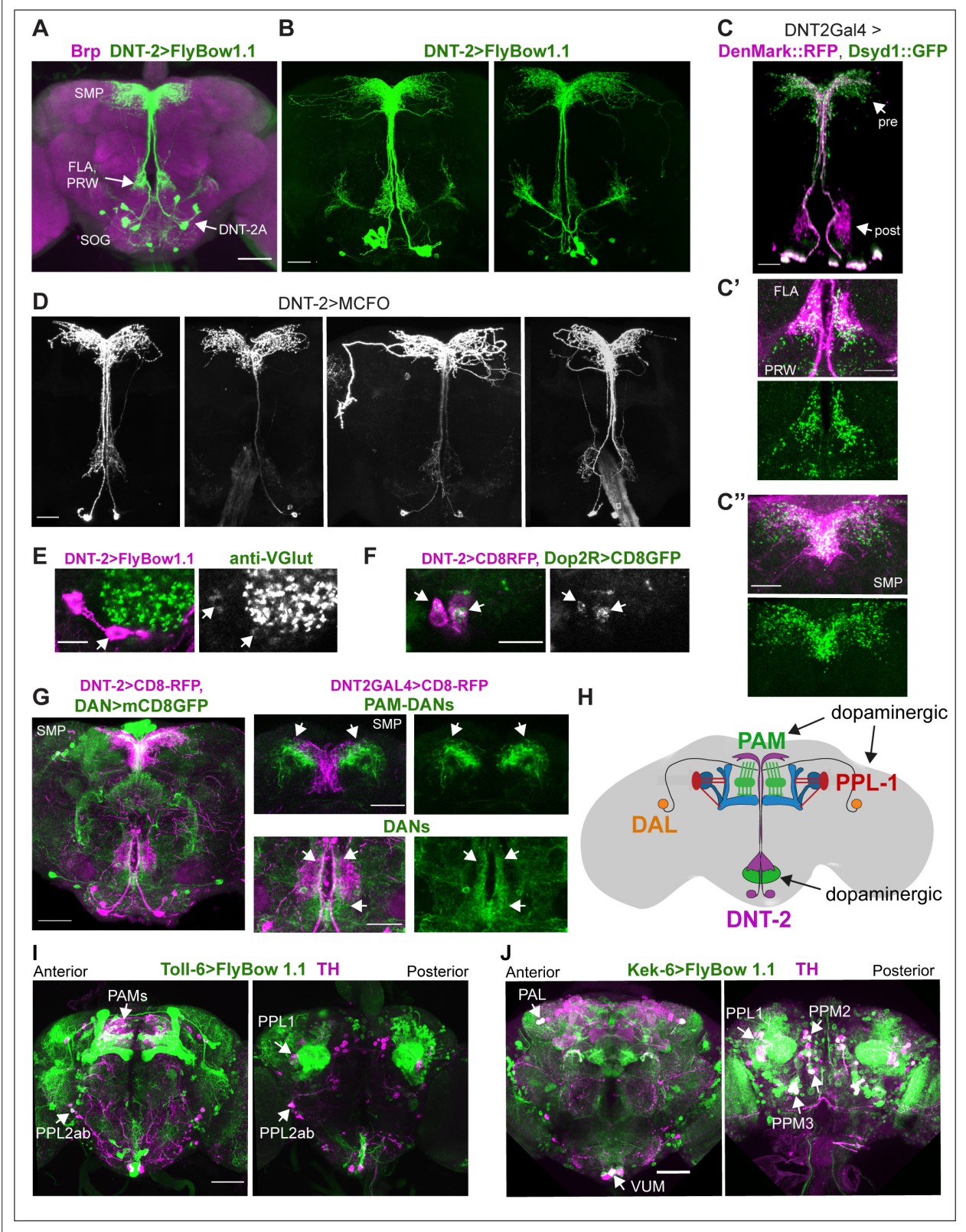

**Figure 1.** Neurons expressing *DNT-2* and its receptors *Toll-6* and *kek-6* in the adult brain. (**A, B**) *DNT-2A*-expressing neurons (*DNT-2>FlyBow1.1* in green; anti-Brp in magenta) have cell bodies in SOG and project to FLA/PRW and SMP. (**C, C', C''**) Pre-synaptic (green) and post-synaptic (magenta) terminals of DNT-2A neurons seen with *DNT-2>DenMark::RFP, Dsyd1::GFP*, higher magnification in (**C', C''**), different specimens from (**C**). DNT-2A projections at SMP and PRW have both pre- and post-synaptic sites. (**D**) Single-neuron DNT-2A>MCFO clones. (**E**) DNT-2A neurons have the vesicular

*Figure 1 continued on next page*

*Figure 1 continued*

glutamate transporter vGlut (arrows). (**F**) Co-localisation between *Dop2RLexA>LexAOP-CD8-GFP* and *DNT2Gal4>UASCD8-RFP* in cell bodies of DNT-2A neurons (arrows). (**G**) Terminals of dopaminergic neurons (*TH>mCD8GFP*) abut and overlap those of DNT-2A neurons (*DNT2>CD8-RFP*, magenta), arrows; magnified projections on the right. (**H**) Illustration of neurons expressing *DNT-2* (magenta) and KCs, DAN PAM and PPL1, and DAL neurons (**I**) *Toll-6>FlyBow1.1* is expressed in Kenyon cells, PPL1, PPL2, and PAM DANs, as revealed by co-localisation with anti-TH. (**J**) *kek-6>FlyBow1.1* co-localises with TH in MB vertical lobes, dopaminergic PALs, VUMs, PPL1, PPM2, and PPM3. SMP: superior medial protocerebrum; PRW: Prow; FLA: Flange; SOG: sub-oesophageal ganglion. Scale bars: (**A, G left, I,** J) 50 μm; (**B, C, C", D, G right**) 30 μm (**C', E, F**) 25 μm. For genotypes and sample sizes, see *Supplementary file 2*.

The online version of this article includes the following figure supplement(s) for figure 1:

**Figure supplement 1.** Identification of neurotransmitter type for DNT-2A neurons.

**Figure supplement 2.** Identification of *Toll-6* and *kek-6*-expressing neurons.

(*Figure 1—figure supplement 1*). DNT-2A terminals overlapped with those of DANs (*Figure 1G*), suggesting they could receive inputs from neuromodulatory neurons. In fact, single-cell RNA-seq revealed transcripts encoding the dopamine receptors *Dop1R1, Dop1R2, Dop2R,* and/or *DopEcR* in DNT-2+ neurons (*Croset et al., 2018*). Using reporters, we found that Dop2R is present in DNT-2A neurons (*Figure 1F*, *Figure 1—figure supplement 1B*), but not Dop1R2 (*Figure 1—figure supplement 1E*). Altogether, these data showed that DNT-2A neurons are glutamatergic neurons that could receive dopaminergic input both at PRW and SMP.

DNT-2 functions via Toll-6 and Kek-6 receptors, and *Toll-6* is expressed in DANs (*McIlroy et al., 2013*). To identify the cells expressing *Toll-6* and *kek-6* and explore further their link to the dopaminergic system, we used *Toll-6Gal4* (*Li et al., 2020*) and *kek-6Gal4* (*Ulian-Benitez et al., 2017*) to drive expression of membrane-tethered *FlyBbow1.1* and assessed their expression throughout the brain. Using anti-tyrosine hydroxilase (TH) – the enzyme that catalyses the penultimate step in dopamine synthesis – to visualise DANs, we found that Toll-6+ neurons included DANs from the PAMs, PPL1, and PPL2 clusters (*Figure 1I*, *Figure 1—figure supplement 2D and F*; *Supplementary file 1*), whilst Kek-6+ neurons included PAM, PAL, PPL1, PPM2, and PPM3 dopaminergic clusters (*Figure 1J*, *Figure 1—figure supplement 2B, E, and F*; *Supplementary file 1*). DNT-2 can also bind various Tolls and Keks promiscuously (*McIlroy et al., 2013*; *Foldi et al., 2017*) and other *Tolls* are also expressed in the dopaminergic system: PAMs express multiple *Toll* receptors (*Figure 1—figure supplement 2F*) and all PPL1s express at least one *Toll* (*Figure 1—figure supplement 2F*). Using MCFO clones revealed that both *Toll-6* and *kek-6* are also expressed in KCs (*Li et al., 2020*; *Figure 1—figure supplement 2I and J*, *Supplementary file 1*), DAL neurons (*Figure 1—figure supplement 2G and H*, *Supplementary file 1*) and MBONs (*Figure 1—figure supplement 2A–C*). In summary, *Toll-6* and *kek-6* are expressed in DANs, DAL, KCs, and MBONs (*Figure 1H*). These cells belong to a circuit required for associative learning, long-term memory, and behavioural output, and DANs are also required for locomotion (*Riemensperger et al., 2011*; *Chen et al., 2012*; *Aso et al., 2014b*; *Boto et al., 2014*; *Adel and Griffith, 2021*; *Huang et al., 2024*). Altogether, our data showed that DNT-2A neurons are glutamatergic neurons that could receive dopamine as they contacted DANs and expressed the *Dop2R* receptor, and that in turn DANs expressed the DNT-2 receptors *Toll-6* and *kek-6*, and therefore could respond to DNT-2. These data suggested that there could be bidirectional connectivity between DNT-2A neurons and DANs, which we explored below.

## Bidirectional connectivity between DNT-2A neurons and DANs

To verify the connectivity of DNT-2A neurons with DANs, we used various genetic tools. To identify DNT-2A output neurons, we used TransTango (*Talay et al., 2017*; *Figure 2A*, *Figure 2—figure supplement 1*). DNT-2A RFP+ outputs included a subset of MB α'β' lobes, αβ KCs, tip of MB β'2, DAL neurons, dorsal fan-shaped body layer, and possibly PAM or other DANs (*Figure 2A*, *Figure 2—figure supplement 1*). Consistently, these DNT-2A output neurons express *Toll-6* and *kek-6* (*Supplementary file 1*). To identify DNT-2A input neurons, we used BAcTrace (*Cachero et al., 2020*). This identified PAM-DAN inputs at SMP (*Figure 2B*). Altogether, these data showed that DNT-2A neurons receive dopaminergic neuromodulatory inputs, their outputs include MB KCs, DAL neurons, and possibly DANs, and DNT-2 arborisations at SMP are bidirectional.

To further test the relationship between DNT-2A neurons and DANs, we reasoned that stimulating DANs would provoke either release or production of dopamine. So, we asked whether increasing

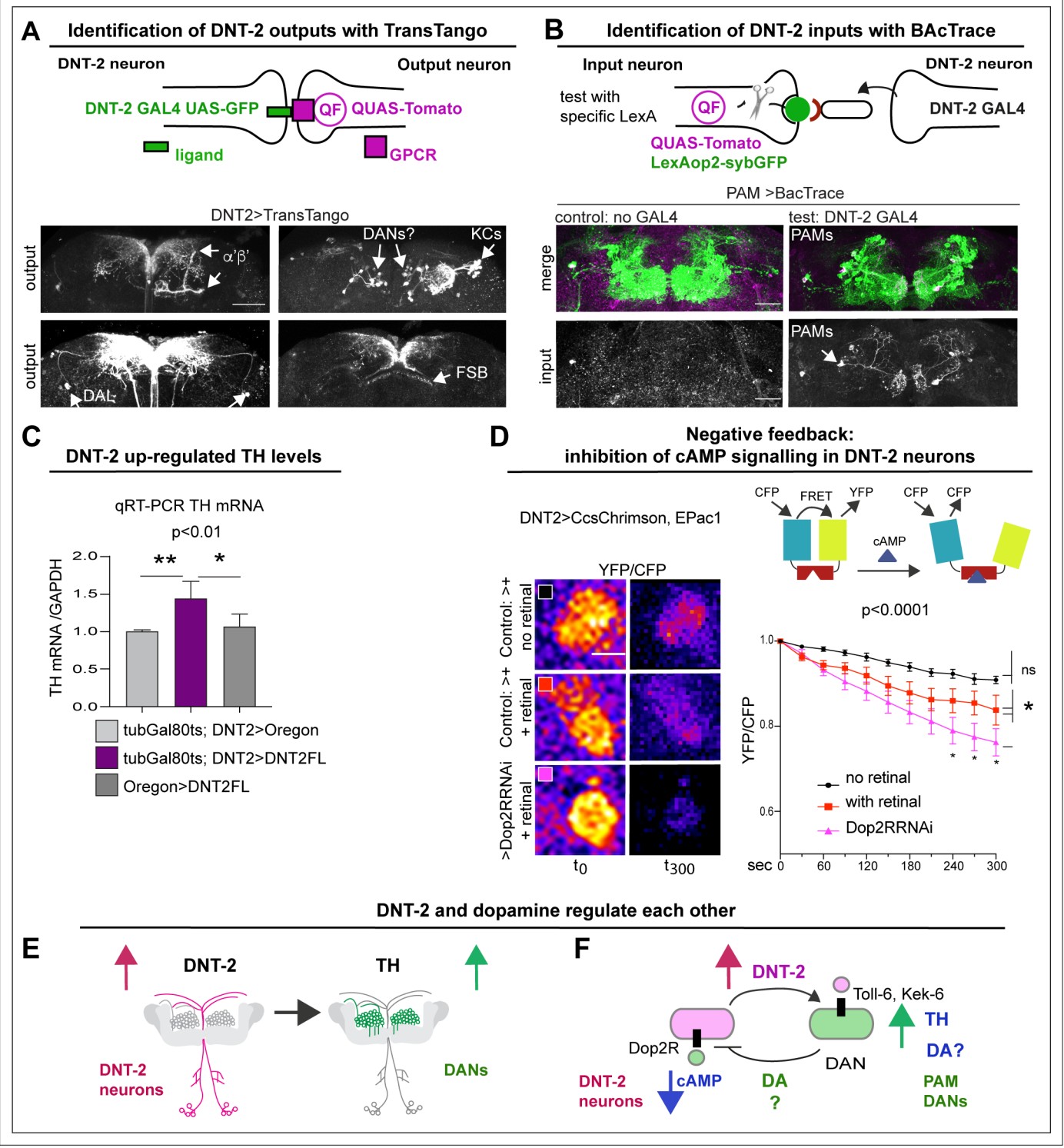

**Figure 2.** DNT-2 neurons are functionally connected to dopaminergic neurons. (**A**) TransTango revealed out-puts of DNT-2 neurons. All neurons express *TransTango* and expression of the TransTango ligand in DNT-2 neurons identified DNT-2 outputs with Tomato (anti-DsRed). TransTango identified as DNT-2 outputs KC α′β′ MB lobes (anterior brain, arrow, top left); Kenyon and possibly DAN cell bodies (posterior, arrows top right); DAL neurons (bottom left, arrows) and the dorsal layer of the fan shaped body (bottom right, arrow). See also *Figure 2—figure supplement 1* for further controls. (**B**) BAcTrace tests connectivity to a candidate neuron input visualised with *LexAop>sybGFP* by driving the expression of a ligand from *DNT-2GAL4* that will activate *QUASTomato* in the candidate input neuron (***Cachero et al., 2020***). Candidate PAM neurons visualised at SMP with GFP (green): Control: *R58E02LexA>BAcTrace 806*, no *GAL4*. Test: *R58E02LexA, DNT-2GAL4>BAcTrace 806* revealed PAMs are inputs of DNT-2A neurons at SMP (Tomato, bottom). Magenta shows QUAS-Tomato. (**C**) qRT-PCR showing that TH mRNA levels increased with DNT-2 over-expression at 30°C (*tubGAL80*[ts], *DNT-*

*Figure 2 continued on next page*

*Figure 2 continued*

*2>DNT-2FL*). One-way ANOVA, p=0.0085; post doc Dunnett's multiple comparison test. Mean ± standard deviation (s.d.). n=4 (left), 5 (middle), 3 (right). (**D**) FRET cAMP probe Epac1 revealed that *DNT-2>Dop2R-RNAi* knock-down decreased YFP/CFP ratio over time in DNT-2A neurons, meaning that cAMP levels increased. Two-way ANOVA, genotype factor p<0.0001, time factor p<0.0001; post doc Dunnett's. Mean ±s.d. n=9,12,17. (**E, F**) Summary: DNT-2 neurons and DANs are functionally connected and modulate each other. (**E**) DNT-2 can induce *TH* expression in DANs; (**F**) this is followed by negative feedback from DANs to DNT-2 neurons (question marks indicate inferences). TH: tyrosine hydroxylase. Scale bars: (**A**) 50 µm; (**B**) 30 µm; (**D**) 20 µm. p-Values over graphs in (**C**) refer to group analyses; stars indicate multiple comparisons tests. *p<0.05, **p<0.01, ***p<0.001. For sample sizes and further statistical details, see *Supplementary file 2*.

The online version of this article includes the following source data and figure supplement(s) for figure 2:

**Source data 1.** Quantitative results used to generate graphs in *Figure 2C*.

**Source data 2.** Quantitative results used to generate graphs in *Figure 2D*.

**Figure supplement 1.** TransTango connectivity controls.

DNT-2 levels in DNT-2 neurons could influence dopamine levels. For this, we over-express *DNT-2* in full-length form (i.e. *DNT-2FL*) as it enables to investigate non-autonomous functions of DNT-2 (*Ulian-Benitez et al., 2017*). Importantly, DNT-2FL is spontaneously cleaved into the mature form (*McIlroy et al., 2013*; *Foldi et al., 2017*) (see 'Discussion'). Thus, we over-expressed *DNT-2FL* in DNT-2 neurons and asked whether this affected dopamine production, using mRNA levels for *TH* as readout. Using quantitative real-time PCR (qRT-PCR), we found that over-expressing *DNT2-FL* in DNT-2 neurons in adult flies increased *TH* mRNA levels in fly heads (*Figure 2C*). This showed that DNT-2 could stimulate dopamine production.

Next, we wondered whether in turn DNT-2A neurons that express *Dop2R* could be modulated by dopamine. Binding of dopamine to D2-like Dop2R (also known as DD2R) inhibits adenylyl-cyclase, decreasing cAMP levels (*Hearn et al., 2002*; *Neve et al., 2004*). Thus, we asked whether DNT-2A neurons received dopamine and signal via Dop2R. Genetic restrictions did not allow us to activate PAMs and test DNT-2 neurons, so we activated DNT-2 neurons and tested whether *Dop2R* knock-down would increase cAMP levels. We used the FRET-based cAMP sensor, Epac1-camps-50A (*Shafer et al., 2008*). When Epac1-camps-50A binds cAMP, FRET is lost, resulting in decreased YFP/CFP ratio over time. Indeed, *Dop2R* RNAi knock-down in DNT-2A neurons significantly increased cAMP levels (*Figure 2D*), demonstrating that normally Dop2R inhibits cAMP signalling in DNT-2A cells. Importantly, this result meant that in controls, activating DNT-2A neurons caused dopamine release from DANs that then bound Dop2R to inhibit adenylyl-cyclase in DNT-2A neurons; this inhibition was prevented with *Dop2R* RNAi knock-down. Altogether, this shows that DNT-2 up-regulated TH levels (*Figure 2E*), and presumably via dopamine release, this inhibited cAMP in DNT-2A neurons (*Figure 2F*).

In summary, DNT-2A neurons are connected to DANs, DAL, and MB KCs, all of which express DNT-2 receptors *Toll-6* and *kek-6* and belong to a dopaminergic as well as associative learning and memory circuit. Furthermore, DNT-2A and PAM neurons form bidirectional connectivity. Finally, DNT-2 and dopamine regulate each other: DNT-2 increased dopamine levels (*Figure 2E*), and in turn dopamine via Dop2R inhibited cAMP signalling in DNT-2A neurons (*Figure 2F*). That is, an amplification was followed by negative feedback. This suggested that a dysregulation in this feedback loop could have consequences for dopamine-dependent behaviours and for circuit remodelling by the DNT-2 growth factor.

## DNT-2 and Toll-6 maintain survival of PAM dopaminergic neurons in the adult brain

We showed earlier that DNT-2 and PAM DANs are connected, so we next asked whether loss of function for *DNT-2* or *Toll-6* would affect PAMs. In wild-type flies, PAM-DAN number can vary between 220 and 250 cells per *Drosophila* brain, making them ideal to investigate changes in cell number (*Liu et al., 2012*). Maintenance of neuronal survival is a manifestation of structural brain plasticity in mammals, where it depends on the activity-dependent release of the neurotrophin BDNF (*Lu et al., 2005*; *Wang et al., 2022*). Importantly, cell number can also change in the adult fly as neuronal activity can induce neurogenesis via Toll-2, whereas DANs are lost in neurodegeneration models (*Feany and Bender, 2000*; *Li et al., 2020*). Thus, we asked whether DNT-2 influences PAM-DAN number in the adult brain. We used *THGal4; R58E02Gal4* to visualise nuclear Histone-YFP in DANs (*Figure 3A*) and

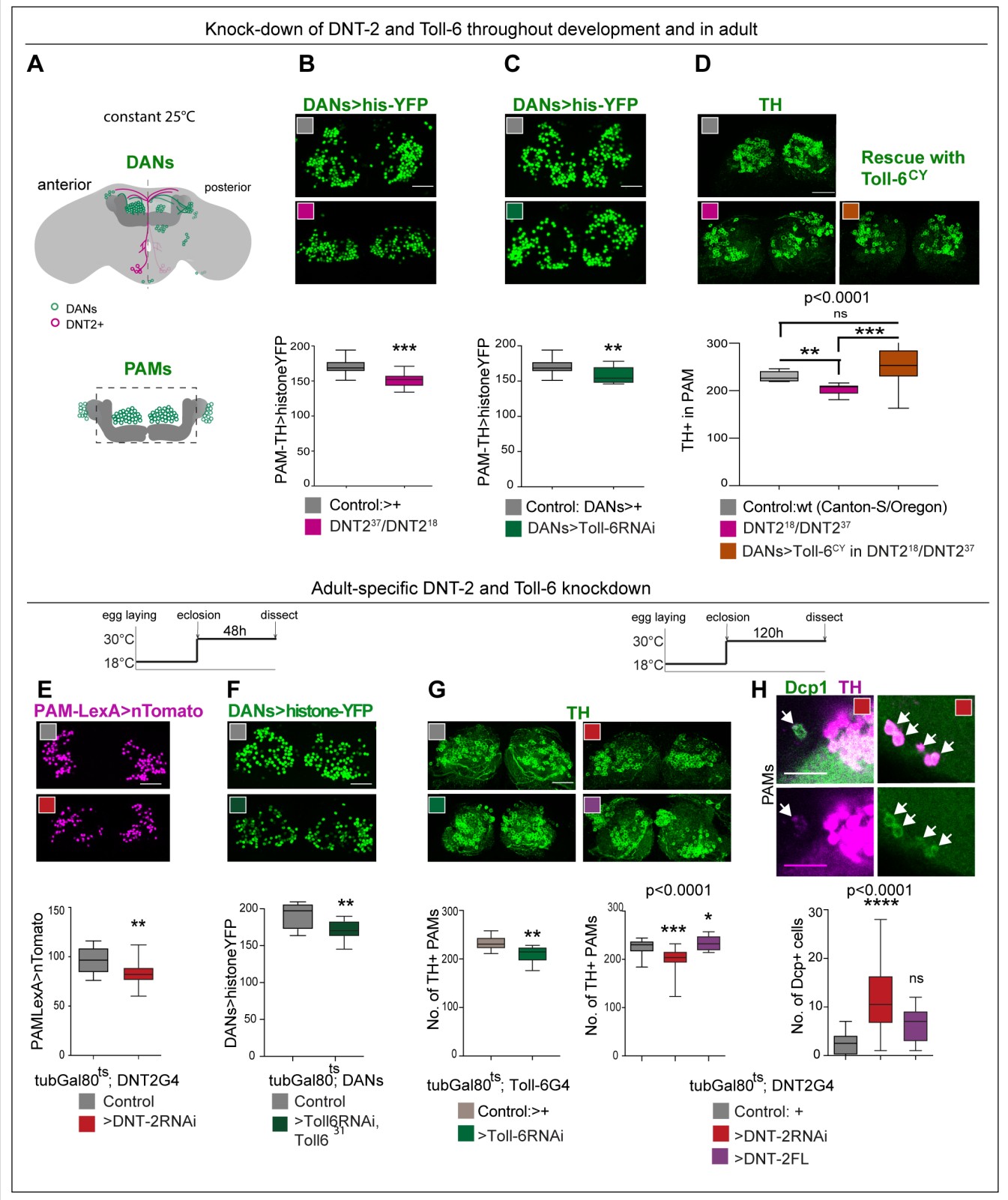

**Figure 3.** DNT-2 and Toll-6 maintain PAM neuron survival in the developing and adult brain. (**A**) Illustration of PAM neuronal cell bodies and experimental temporal profile. DANs are shown in green, DNT-2A neurons in magenta, and MB in dark grey. The left hemisphere shows the anterior brain with PAL and PAM DAN neurons (green) and DNT-2 neurons (magenta); the right shows the posterior brain, with the calyx and other DAN neurons (PPM1, PPM2, PPM3, PPL1, PPL2, green). (**B–D**) Fruit flies were kept constantly at 25°C, from development to adult. Analyses done in adult brains. (**B**)

*Figure 3 continued on next page*

*Figure 3 continued*

$DNT-2^{37}/DNT-2^{18}$ mutants had fewer histone-YFP+-labelled PAM neurons (*THGAL4, R58E02-GAL4>hisYFP*). Unpaired Student's *t*-test. (**C**) *Toll-6* RNAi knock-down in all DANs (*THGAL4, R58E02-GAL4>hisYFP, Toll-6 RNAi*) reduced Histone-YFP+-labelled PAM cell number. Unpaired Student's *t*-test. (**D**) $DNT-2^{37}/DNT-2^{18}$ mutants had fewer PAMs stained with anti-TH antibodies. Over-expressing *Toll-6^{CY}* in DANs (*THGAL4, R58E02 GAL4>hisYFP, Toll-6^{CY}*) rescued TH+ PAM neurons in $DNT-2^{37}/DNT-2^{18}$ mutants, demonstrating that DNT-2 functions via Toll-6 to maintain PAM cell survival. Welch ANOVA p<0.0001, post hoc Dunnett test. (**E–H**) Adult-specific restricted over-expression or knock-down at 30°C using the temperature-sensitive GAL4 repressor *tubGAL80^{ts}*. (**E**) Adult-specific *DNT-2* RNAi knock-down in DNT-2 neurons decreased Tomato+ PAM cell number (*tubGAL80^{ts}, R58E02-LexA, DNT-2 GAL4>LexAOP-Tomato, UAS DNT-2-RNAi*). Unpaired Student's *t*-test, p=0.005. (**F**) Adult-specific *Toll-6* RNAi knock-down in *Toll-6^{31}* heterozygous mutant flies in DANs, reduced Histone-YFP+ PAM cell number (*tubGAL80^{ts}; THGAL4, R58E02-GAL4>hisYFP, Toll-6 RNAi/Toll6^{31}*). Unpaired Student's *t*-test. (**G**) PAMs were visualised with anti-TH. Left: *tubGAL80^{ts}, Toll-6>Toll-6-RNAi* knock-down decreased TH+ PAM cell number. Unpaired Student's *t*-test. Right: *tubGAL80^{ts}, DNT-2>DNT-2-RNAi* knock-down decreased TH+ PAM cell number, whereas *DNT-2FL* over-expression increased PAM cell number. Kruskal–Wallis ANOVA, p=0.0001, post hoc Dunn's test. (**H**) Adult-specific *tubGAL80^{ts}, DNT-2>DNT-2RNAi* knock-down increased the number of apoptotic cells in the brain labelled with anti-DCP-1. Dcp-1+ cells co-localise with anti-TH at least in PAM clusters. One-way ANOVA, p<0.0001, post hoc Bonferroni's multiple comparisons test. DANs>histone-YFP: all dopaminergic neurons expressing histone-YFP, genotype: *THGal4 R58E02Gal4>UAS-histoneYFP*. *PAMsLexA>tomato*: restricted to PAM DANs: *R58E02LexA >LexOp-nlstdTomato*. Controls: *GAL4* drivers crossed to wild-type Canton-S. Scale bars: (**B–G**) 30 μm; (**H**) 20 μm. Graphs show boxplots around the median. p-values over graphs in (**D, G** right, **H**) refer to group analyses; stars indicate multiple comparisons tests. *p<0.05, **p<0.01, ***p<0.001. For further genotypes, sample sizes, and statistical details, see *Supplementary file 2*.

The online version of this article includes the following source data for figure 3:

**Source code 1.** DeadEasy Central Brain was used to automatically count the number of PAM neuron nuclei labelled with His-YFP in adult brains.

**Source code 2.** DeadEasy Dopaminergic-3D was developed and used to automatically or semi-automatically count the number of PAM neurons stained with anti-TH antibodies in adult brains.

**Source code 3.** Cells filter, to function together with DeadEasy Dopaminergic-3D.

**Source data 1.** Quantitative results used to generate graphs in *Figure 3B*.

**Source data 2.** Quantitative results used to generate graphs in *Figure 3C*.

**Source data 3.** Quantitative results used to generate graphs in *Figure 3D*.

**Source data 4.** Quantitative results used to generate graphs in *Figure 3E*.

**Source data 5.** Quantitative results used to generate graphs in *Figure 3F*.

**Source data 6.** Quantitative results used to generate graphs in *Figure 3G*.

**Source data 7.** Quantitative results used to generate graph in *Figure 3H*.

counted automatically YFP+ PAMs using a purposely modified DeadEasy plug-in developed for the adult fly brain (*Li et al., 2020*). DeadEasy plug-ins were developed and used before to count cells labelled with sparsely distributed nuclear markers in embryos (*Zhu et al., 2008*; *Forero et al., 2009*; *Forero et al., 2010a*; *Forero et al., 2010b*; *McIlroy et al., 2013*), larvae (*Kato et al., 2011*; *Forero et al., 2012*; *Losada-Perez et al., 2016*), and adult (*Li et al., 2020*) *Drosophila* brains. Here, we show that $DNT2^{37}/DNT2^{18}$ mutant adult brains had fewer PAMs than controls (*Figure 3B*). Similarly, *Toll-6* RNAi knock-down in DANs also decreased PAM neuron number (*Figure 3C*). DAN loss was confirmed with anti-TH antibodies and counted manually as there were fewer TH+ PAMs in $DNT2^{37}/DNT2^{18}$ mutants (*Figure 3D*). Importantly, PAM cell loss was rescued by over-expressing activated *Toll-6^{CY}* in DANs in *DNT-2* mutants (*Figure 3D*). Altogether, these data showed that DNT-2 functions via Toll-6 to maintain PAM neuron survival.

To ask whether DNT-2 could regulate DAN number specifically in the adult brain, we used *tubGal80^{ts}* to conditionally knock-down gene expression in the adult. PAMs were visualised with either *R58E02LexA>LexAop-nls-tdTomato* or *THGal4; R58E02Gal4>histone-YFP* and counted automatically. Adult-specific *DNT-2* RNAi knock-down decreased Tomato+ PAM cell number (*Figure 3E*). Similarly, RNAi *Toll-6* knock-down in DANs also decreased PAM neuron number (*Figure 3F*). Furthermore, knock-down of either *Toll-6* or *DNT-2* in the adult brain caused loss of PAM neurons visualised with anti-TH antibodies and counted manually (*Figure 3G*). Cell loss was due to cell death as adult-specific *DNT-2 RNAi* knock-down increased the number of apoptotic cells labelled with anti-*Drosophila* Cleave caspase-1 (DCP-1) antibodies compared to controls, including Dcp-1+ PAMs and other TH+ cells (*Figure 3H*). Dcp-1+ cells also included TH-negative cells, consistent with the expression of *Toll-6* and *kek-6* also in other cell types. In contrast, *DNT-2FL* over-expression in DNT2 neurons did not alter the incidence of apoptosis (*Figure 3G*), consistently with the fact that DNT-2FL spontaneously cleaves into the mature form (*McIlroy et al., 2013*; *Foldi et al., 2017*). Instead, and importantly, over-expression

of DNT-2FL increased PAM cell number (*Figure 3G*). Thus, *DNT-2* and *Toll-6* knock-down specifically in the adult brain induced apoptosis and PAM-neuron loss, whereas DNT-2 gain of function increased PAM cell number.

Altogether, these data showed that PAM cell number is plastic, sustained PAM neuron survival in development, and in the adult brain depends on DNT-2 and Toll-6, and a reduction in their levels causes DAN cell loss, characteristic of neurodegeneration.

## DNT-2 and its receptors are required for arborisations and synapse formation

We next asked whether DNT-2, Toll-6, and Kek-6 could influence dendritic and axonal arbours and synapses of DANs (*Figure 4A*). Visualising the pre-synaptic reporter Synaptotagmin-GFP (Syt-GFP) in all DANs (with *THGal4; R58E02Gal4*), we found that *DNT-2^18^/DNT-2^37^* mutants completely lacked DAN synapses in the MBβ,β' and γ lobes (*Figure 4B*). Interestingly, DAN connections at α,α' lobes were not affected (*Figure 4B*). This shows that DNT-2 is required for synaptogenesis and connectivity of DANs to MB β,β' and γ lobes.

PAM-β2β'2 neuron dendrites overlap axonal DNT2 projections. *Toll-6* RNAi knock-down in PAM-β2β'2 (with split-GAL4 *MB301BGal4*; *Aso et al., 2014a*) reduced dendrite complexity (*Figure 4C*). To test whether DNT-2 could alter these dendrites, we over-expressed mature *DNT-2CK*. DNT-2CK is not secreted (from transfected S2 cells), but it is functional in vivo (*Zhu et al., 2008*; *Foldi et al., 2017*; *Ulian-Benitez et al., 2017*). Importantly, over-expressed *DNT-2CK* functions cell-autonomously, whereas *DNT-2FL* functions also non-autonomously, but they have similar effects (*Zhu et al., 2008*; *Foldi et al., 2017*; *Ulian-Benitez et al., 2017*). Over-expression of *DNT-2CK* in PAM-β2β'2 increased dendrite arbour complexity (*Figure 4D*). Thus, DNT-2 and its receptor Toll-6 are required for dendrite growth and complexity in PAM neurons.

To ask whether DNT-2 could affect axonal terminals, we tested PPL1 axons. PPL1-γ1-pedc neurons have a key function in long-term memory gating (*Aso et al., 2012*; *Plaçais et al., 2012*; *Aso et al., 2014a*; *Boto et al., 2020*; *Huang et al., 2024*) and express both *Toll-6* and *kek-6* (*Supplementary file 1*). Using split-GAL4 line M*B320C-Gal4* to visualise PPL1 axonal arbours, RNAi knock-down of either *Toll-6*, *kek-6* or both together caused axonal misrouting away from the mushroom body peduncle (*Figure 4E and E'*, chi-square p<0.05, see *Supplementary file 2*). Similarly, *DNT-2FL* over-expression also caused PPL1 misrouting (*Figure 4F*, chi-square p<0.05, see *Supplementary file 2*). Thus, DNT-2, Toll-6, and Kek-6 are required for appropriate targeting and connectivity of PPL1 DAN axons.

To test whether this signalling system was required specifically in the adult brain, we used *tubGAL80^ts^* to knock-down *Toll-6* and *kek-6* with RNAi conditionally in the adult and visualised the effect on synaptogenesis using the post-synaptic reporter Homer-GCaMP and anti-GFP antibodies. Adult-specific *Toll-6 kek-6* RNAi knock-down in PAM neurons did not affect synapse number (not shown), but it decreased post-synaptic density (PSD) size, both at the MB lobe and at the SMP dendrite (*Figure 4G*). These data meant that the DNT-2 receptors *Toll-6* and *kek-6* continue to be required in the adult brain for appropriate synaptogenesis.

Altogether, these data showed that DNT-2, Toll-6, and Kek-6 are required for dendrite branching, axonal targeting, and synapse formation. The shared phenotypes from altering the levels of *DNT-2* and *Toll-6 kek-6* in arborisations and synapse formation support their joint function in these contexts. Importantly, these findings showed that the connectivity of PAM and PPL1 DANs depends on DNT-2, Toll-6, and Kek-6.

## DNT-2 neuron activation and *DNT-2* over-expression induced synapse formation in target PAM dopaminergic neurons

The above data showed that DNT-2, Toll-6, and Kek-6 are required for DAN cell survival, arborisations, and synaptogenesis in development and adults. This meant that the dopaminergic circuit remains plastic in adult flies, consistently with their functional plasticity (*Boto et al., 2014*). Thus, we wondered whether neuronal activity could also induce remodelling in PAM neurons. In mammals, neuronal activity induces translation, release, and cleavage of BDNF, and BDNF drives synaptogenesis (*Poo, 2001*; *Lu et al., 2005*; *Lu et al., 2013*; *Wang et al., 2022*). Thus, we first asked whether neuronal activity could influence DNT-2 levels or function. We visualised tagged DNT-2FL-GFP in adult brains, activated DNT-2 neurons with TrpA1 at 30°C, and found that DNT-2 neuron activation increased the number of

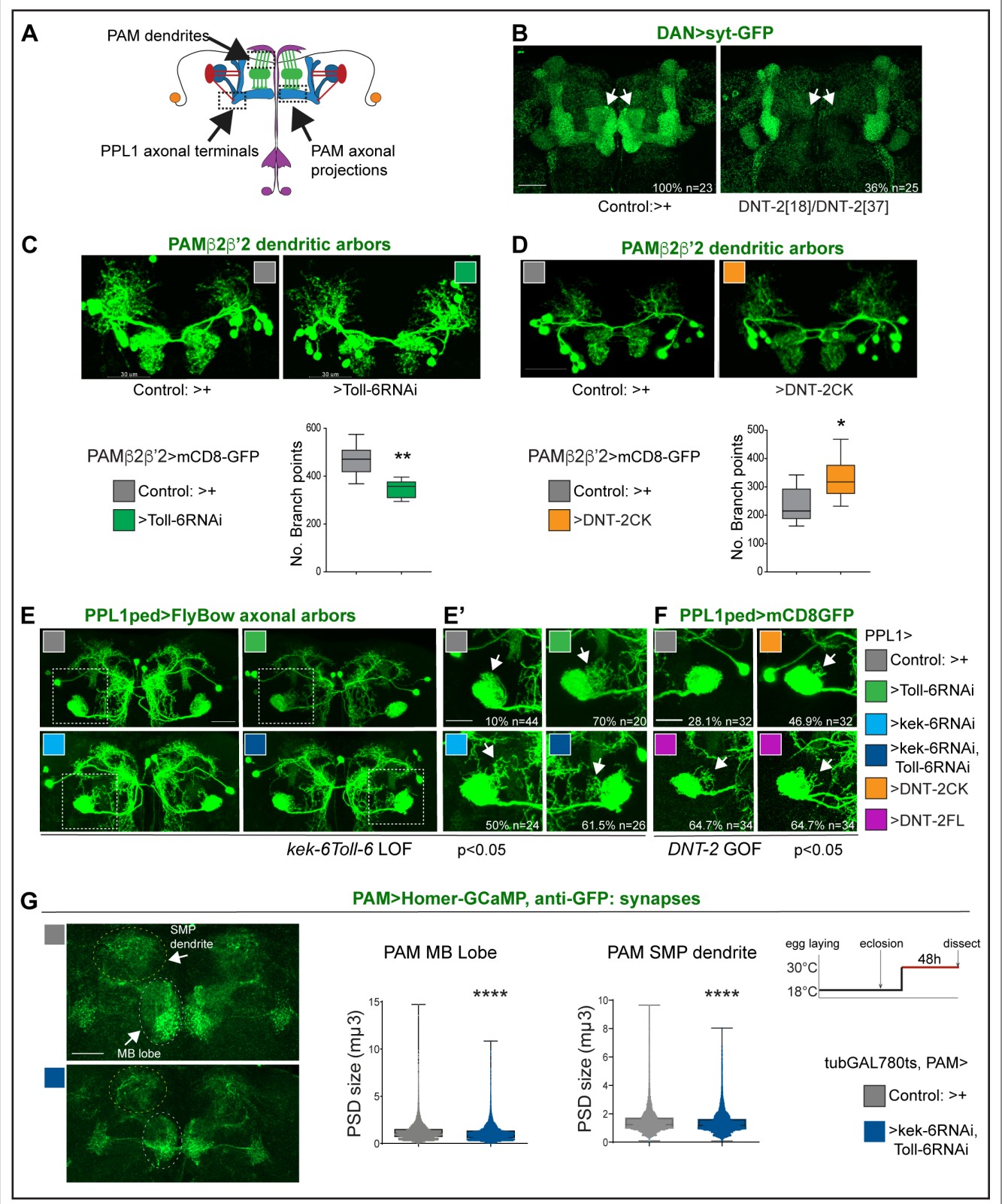

**Figure 4.** DNT-2, Toll-6, and Kek-6 are required for arborisations and synapse formation. (**A**) Illustration showing the regions of interest (ROIs) (dashed lines) used for the analyses, corresponding to dendrites and axonal endings of PAMs and axonal terminals of PPL1ped neurons. (**B–F**) Fruit flies were kept constantly at 25°C. (**B**) Complete loss of DAN synapses (*THGAL4, R58E02GAL4>syt-GFP*) onto α,β MB lobe in *DNT-2[37]/DNT-2[18]* null mutants. (**C**) *Toll-6* RNAi knock-down in PAM β2β'2 neurons (*MB301B>FB1.1, Toll-6-RNAi*) decreased dendrite complexity. Unpaired Student's *t*-test. (**D**) Over-

*Figure 4 continued on next page*

*Figure 4 continued*

expression of cleaved *DNT-2CK* in PAM β2β'2 neurons (*MB301B>CD8-GFP, DNT-2CK*) increased dendrite complexity. Unpaired Student's *t*-test. Same magnification as (C). (**E, E', F**) PPL1ped axonal misrouting was visualised with *split-GAL4 MB320CGal4>FlyBow1.1*. Images show PPL1-γped neurons and some PPL1-α2α'2. (**E,E'**) RNAi knock-down of *Toll-6, kek-6* or both (e.g. *MB320CGal4>FlyBow1.1, Toll-6RNAi*) in PPL1-γped neurons caused axonal terminal misrouting (arrows). (**E'** Higher magnification of **E**, dotted squares). Chi-square for group analysis: p=0.0224, and multiple comparisons Bonferroni correction control vs *Toll-6RNAi* p<0.01; vs *kek-6RNAi* p<0.05; vs *Toll-6RNAi kek-6RNAi* p<0.01, see ***Supplementary file 2***. (**F**) PPL1 misrouting was also induced by over-expressed *DNT-2CK* or *DNT-2FL* (e.g. *MB320CGal4>FlyBow1.1, DNT-2FL*). Chi-square for group analysis: p<0.05, Bonferroni correction control vs *DNT-2CK* ns, vs *DNT-2FL* *p<0.05, see ***Supplementary file 2***. (**G**) Adult-specific *Toll-6 kek-6 RNAi* knock-down in PAM neurons decreased size of post-synaptic density sites (*PAM >Homer-GCaMP* and anti-GFP antibodies). Temperature regime shown on the right. (**C, D**) Graphs show boxplots around the median; (**G**) are boxplots with dot plots. (**C, D, G**) *p<0.05, **p<0.01, ****p<0.0001. Scale bars: (**B, C, D, E**) 30 μm; (**E', F, G**) 20 μm. For genotypes, sample sizes, and statistical details, see ***Supplementary file 2***.

The online version of this article includes the following source data for figure 4:

**Source data 1.** Quantitative results used to generate graphs in ***Figure 4B***.

**Source data 2.** Quantitative results used to generate graphs in ***Figure 4C***.

**Source data 3.** Quantitative results used to generate graphs in ***Figure 4D***.

**Source data 4.** Quantitative results used to generate graphs in ***Figure 4E***.

**Source data 5.** Quantitative results used to generate graphs in ***Figure 4F***.

**Source data 6.** Quantitative results used to generate graphs in ***Figure 4G***.

DNT-2-GFP vesicles produced (***Figure 5—figure supplement 1A***). Furthermore, neuronal activity also facilitated cleavage of DNT-2 into its mature form. In western blots from brains over-expressing *DNT-2FL-GFP*, the levels of full-length DNT-2FL-GFP were reduced following neuronal activation and the cleaved DNT-2CK-GFP form was most abundant (***Figure 5—figure supplement 1B***). These findings meant that, like mammalian BDNF, DNT-2 can also be influenced by activity.

Thus, we asked whether neuronal activity and DNT-2 could influence synapse formation. We first tested DNT-2 neurons. Activating DNT-2 neurons altered DNT-2 axonal arbours (***Figure 5A***) and it increased Homer-GFP+ synapse number in the DNT-2 SMP arbour (***Figure 5B***, ***Figure 5—figure supplement 2***). Next, as DNT-2 and PAMs form bidirectional connexions at SMP (***Figures 1 and 2***), we asked whether activating DNT-2 neurons could affect target PAM neurons. To manipulate DNT-2 neurons and visualise PAM neurons concomitantly, we combined *DNT-2GAL4* with the *PAM-LexA* driver. However, there were no available *LexA/OP* post-synaptic reporters, so we used the pre-synaptic *LexAOP-Syt-GCaMP* reporter instead, which labels Synaptotagmin (Syt), and GFP antibodies. Activating DNT-2 neurons with TrpA1 increased the number of Syt+ synapses at the PAM SMP arbour (***Figure 5C***) and reduced their size (***Figure 5C***). This was consistent with the increase in Homer-GFP+ PSD number in stimulated DNT-2 neurons (***Figure 5B***). Neuronal activity can induce ghost boutons, immature synapses that are later eliminated (***Fuentes-Medel et al., 2009***). Here, the coincidence of increased pre-synaptic Syt-GFP from PAMs and post-synaptic Homer-GFP from DNT-2 neurons at SMP suggests that newly formed synapses could be stable. PAM neurons also send an arborisation at the MB β, β', γ lobes, but DNT-2 neuron activation did not affect synapse number nor size there (***Figure 5C***). These data showed that activating DNT-2 neurons induced synapse formation at the SMP connection with PAMs.

Finally, we asked whether, like activity, DNT-2FL could also drive synaptogenesis. We over-expressed *DNT-2FL* in DNT-2 neurons and visualised the effect in PAM neurons. Over-expression of *DNT-2FL* in DNT-2 neurons did not alter Syt+ synapse number at the PAM SMP dendrite, but it increased bouton size (***Figure 5D***). In contrast, at the MB β, β' lobe arborisation, over-expressed *DNT-2* did not affect Syt+ bouton size, but it increased the number of output synapses (***Figure 5D***). This data showed that DNT-2 released from DNT-2 neurons could induce synapse formation in PAM target neurons.

Altogether, these data showed that neuronal activity induced synapse formation, stimulated production and cleavage of DNT-2, and DNT-2 could induce synapse formation in target neurons.

## Structural plasticity by DNT2 modified dopamine-dependent behaviour

Circuit structural plasticity raises the important question of what effect it could have on brain function, that is, behaviour. The above data showed that DANs and DNT-2 neurons are functionally connected, loss of function for *DNT-2* or its receptors caused DAN loss, altered DAN arborisations and caused

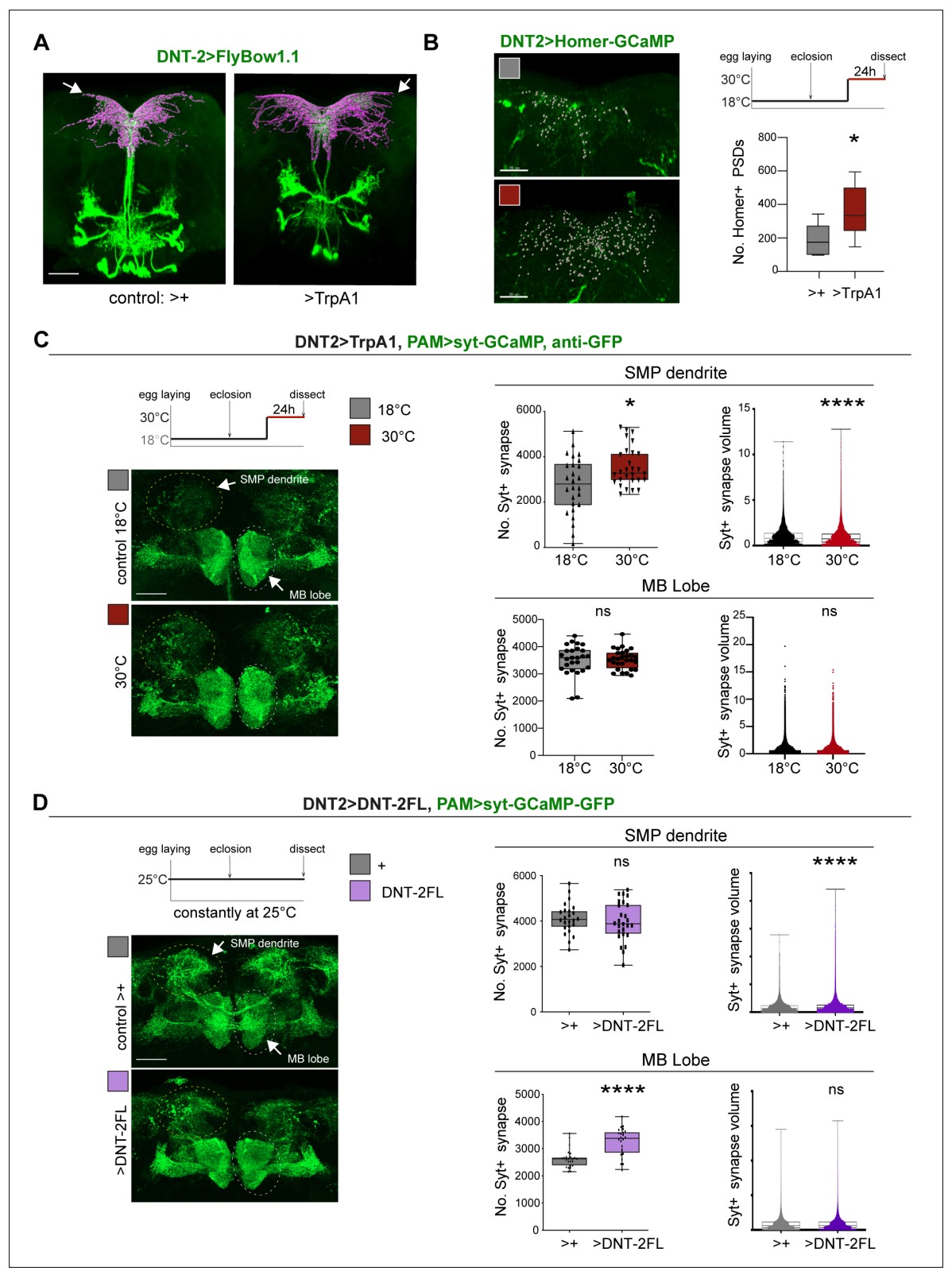

**Figure 5.** DNT-2 neuron activation and DNT-2 over-expression induced synaptogenesis. (**A**) Thermo-activation of DNT-2 neurons at 30°C altered DNT-2 arborisations, here showing an example with smaller dendrites and enlarged axonal arbours (magenta shows 3D rendering of axonal arborisation done with Imaris, merged with raw image in green) (genotype: *DNT-2>FlyBow1.1, TrpA1*). (**B**) Thermogenetic activation of DNT-2 neurons increased the number of Homer+ PSDs in DNT-2 neurons, at SMP (*DNT-2>Homer-GCAMP3, TrpA1*, anti-GFP). Test at 30°C 24 hr: unpaired Student's *t*-test. See

*Figure 5 continued on next page*

*Figure 5 continued*

**Supplementary file 2**. (C) Thermogenetic activation of DNT-2 neurons induced synaptogenesis in PAM target neurons at SMP, but not at MB lobe (genotype: *PAM(R58E02)LexA/LexAOP-sytGCaMP; DNT-2GAL4/UASTrpA1*). At SMP: No. Syt+ synapses: Mann–Whitney *U*; Syt+ synapse volume: Mann–Whitney *U*. At MB lobe: No. Syt+ synapses: unpaired Student's *t* ns; Syt+ synapse volume: Mann–Whitney *U* ns. (D) Over-expression of *DNT-2FL* in DNT-2 neurons increased synapse volume at SMP dendrite and induced synaptogenesis at MB lobe (genotype: *PAM(R58E02)LexA/LexAopSytGcaMP6; DNT-2Gal4/UAS-DNT-2FL*). At SMP: No. Syt+ synapses: unpaired Student's *t* ns. Syt+ synapse volume: Mann–Whitney *U*. At MB lobe: No. Syt+ synapses: unpaired Student's *t*; Syt+ synapse volume: Mann–Whitney *U* ns. Graphs show boxplots around the median, except for PSD volume data that are dot plots. *p<0.05, ****p<0.0001; ns: not significantly different from control. Scale bars (**A-D**): 30 μm. For genotypes, sample sizes, p-values, and other statistical details, see **Supplementary file 2**.

The online version of this article includes the following source data and figure supplement(s) for figure 5:

**Source data 1.** Quantitative results used to generate graphs in **Figure 5B**.

**Source data 2.** Quantitative results used to generate graphs in **Figure 5C**.

**Source data 3.** Quantitative results used to generate graphs in **Figure 5D**.

**Figure supplement 1.** Neuronal activity increased production and cleavage of DNT-2.

**Figure supplement 1—source data 1.** Quantitative data used to generate graph in **Figure 5—figure supplement 1A**.

**Figure supplement 1—source data 2.** Images of the original western blot used in **Figure 5—figure supplement 1B** to indicate (left) the positions of the molecular weight ladders and (right) labelling on the original test membrane with DNT-2-FL-GFP and its cleavage with activity (TrpA1 at 30°C).

**Figure supplement 1—source data 3.** Original western blot membrane showing DNT-2FL-GFP activity-dependent cleavage.

**Figure supplement 2.** Controls for **Figure 5B**.

**Figure supplement 2—source data 1.** Quantitative data used to generate graph in **Figure 5—figure supplement 2**.

---

synapse loss or reduction in size, and DNT-2 could induce neuron number, dendrite branching and synaptogenesis, altogether modifying circuit connectivity. To measure the effect of such circuit modifications on brain function, we used dopamine-dependent behaviours as readout.

Startle-induced negative geotaxis (also known as the climbing assay) is commonly used as a measure of locomotor ability and requires dopamine and specifically PAM neuron function (*Riemensperger et al., 2013*; *Sun et al., 2018*). We tested the effect of *DNT-2* or *Toll-6* and *kek-6* loss of function in climbing, and both *DNT-2³⁷/DNT-2¹⁸* mutants and flies in which *DNT-2* was knocked-down in DNT-2 neurons in the adult stage had lower climbing ability than controls (*Figure 6A*). Similarly, when *Toll-6* and *kek-6* were knocked-down with RNAi in the adult using a *Toll-6-* or a *PAM-GAL4* neuron driver, climbing was also reduced (*Figure 6B*). Importantly, over-expressing activated *Toll-6^{CY}* in DANs rescued the locomotion deficits of *DNT-2* mutants, showing that DNT-2 functions via Toll-6 in this context (*Figure 6C*).

We also tracked freely moving flies in an open arena (*Eyjolfsdottir et al., 2014*). Interestingly, in that setting, locomotion of homozygous *DNT-2³⁷/DNT-2¹⁸* mutants was similar to that of controls, but over-expression of *Toll-6^{CY}* in their DANs increased locomotion as flies walked longer distances and spent less time immobile (*Figure 6D*). Adult flies over-expressing *DNT2-FL* walked faster (*Figure 6E*, *Figure 5—figure supplement 2*) and so did those where DNT-2 neurons were activated with TrpA1 (*Figure 6F*, *Figure 6—figure supplement 1*), consistently with the fact that neuronal activity increased DNT-2 production (*Figure 5—figure supplement 1A*) and that DNT-2FL increased TH levels (*Figure 2C*). Therefore, increased Toll-6^{CY} levels in DANs increase locomotion, and increased DNT-2 levels are sufficient to boost walking speed. Interestingly, both loss and gain of function for *DNT-2* also caused seizures (*Figure 6—figure supplement 2*). Thus, dopamine-dependent locomotion is regulated by the function of DNT-2, Toll-6, and Kek-6.

Next, as dopamine is an essential neurotransmitter for learning and memory (*Adel and Griffith, 2021*), we asked whether DNT-2 might influence appetitive olfactory conditioning. Starved flies were trained to associate a sugar reward with an odour (CS+) while another odour was presented without sugar (CS-), and their preference for CS+ versus CS- was measured, 24 hr after training (*Tempel et al., 1983*; *Krashes and Waddell, 2008*; *Krashes and Waddell, 2011*). Remarkably, over-expression of *DNT-2FL* in DNT-2 neurons in adults enhanced appetitive long-term memory (*Figure 6G*), consistent with the positive role of DNT-2 in synaptogenesis that we demonstrated above.

In summary, we have shown that alterations in DNT-2, Toll-6, and Kek-6 levels that caused structural phenotypes in DANs also modified dopamine-dependent behaviours, locomotion, and long-term memory.

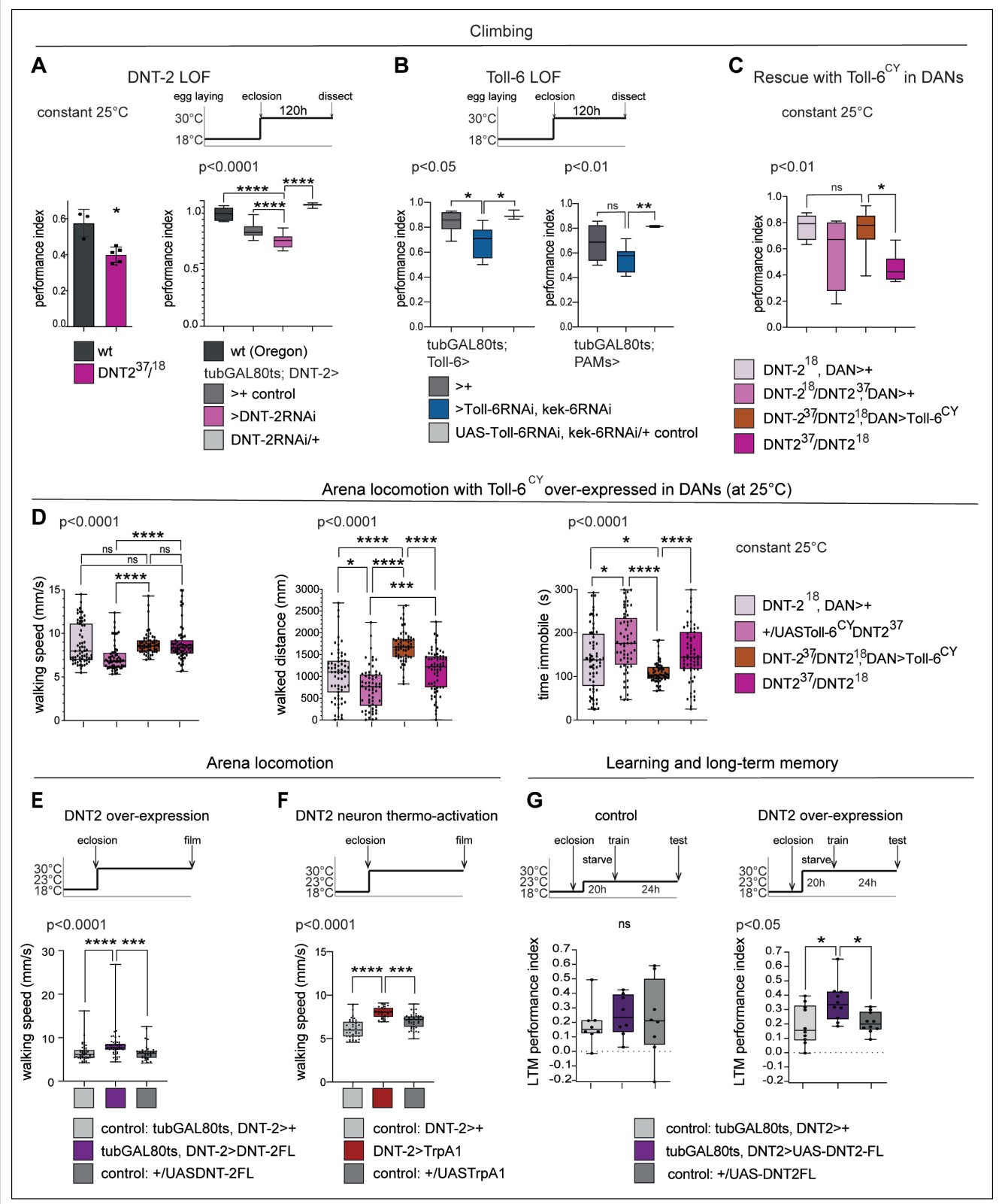

**Figure 6.** DNT-2-induced circuit plasticity modified dopamine-dependent behaviour. (**A**) *DNT-2* mutants (left, *DNT-2³⁷/DNT-2¹⁸*) and flies with adult-specific RNAi knock-down of *DNT-2* in DNT-2 neurons (right, *tubGAL80ᵗˢ; DNT-2>DNT-2RNAi*), had impaired climbing. Left: Mann–Whitney *U*; right: one-way ANOV, post hoc Dunnett. (**B**) Adult-specific *Toll-6* and *kek-6* RNAi knock-down in Toll-6 (*tubGAL80ᵗˢ; Toll-6>Toll-6RNAi, kek-6RNAi,* left) or PAM (*tubGAL80ᵗˢ; R58E02>Toll-6RNAi, kek-6RNAi,* right) neurons impaired climbing. Left: Welch ANOVA, post hoc Dunnett. Right: Welch ANOVA,

*Figure 6 continued on next page*

*Figure 6 continued*

post hoc Dunnett. (**C**) The climbing impairment of *DNT-2* mutants could be rescued with the over-expression of *Toll-6$^{CY}$* in dopaminergic neurons (rescue genotype: UASToll-6$^{CY}$/+; DNT-2$^{18}$THGAL4 R58E02GAL4/DNT-2$^{37}$). Welch ANOVA, post hoc Dunnett. (**D**) Adult-specific over-expression of *Toll-6$^{CY}$* in DANs increased locomotion in *DNT-2$^{37}$/DNT-2$^{18}$* mutants (test genotype: *UASToll-6$^{CY}$/+; DNT-2$^{18}$THGAL4 R58E02GAL4/DNT-2$^{37}$*). Walking speed: Kruskal–Wallis ANOVA, post hoc Dunn's. Distance walked: Kruskal–Wallis ANOVA, post hoc Dunn's. Time spent immobile: Kruskal–Wallis ANOVA, post hoc Dunn's. (**E**) Adult-specific *DNT-2FL* overexpression in DNT-2 neurons increased fruit fly locomotion speed in an open arena at 30°C (see also *Figure 6—figure supplement 1A* for further controls) (test genotype: *tubGAL80$^{ts}$, DNT-2>DNT-2FL*) Kruskal–Wallis, post hoc Dunn's test. (**F**) Thermogenetic activation of DNT-2 neurons at 30°C (*DNT-2>TrpA1*) increased fruit fly locomotion speed (see also *Figure 6—figure supplement 1B* for further controls). One-way ANOVA p<0.0001, post hoc Dunnett's test. (**G**) Over-expression of *DNT-2-FL* in DNT-2 neurons increased long-term memory (test genotype: *tubGAL80$^{ts}$, DNT-2>DNT-2FL*). Left: 23°C controls: one-way ANOVA p=0.8006. Right: 30°C: one-way ANOVA, post hoc Dunnett's test. Graphs show boxplots around the median, under also dot plots. p-Values over graphs refer to group analyses; asterisks indicate multiple comparisons tests. *p<0.05, **p<0.01, ***p<0.001, ****p<0.0001; ns, not significantly different from control. For further genotypes, sample sizes, p-values, and other statistical details, see *Supplementary file 2*.

The online version of this article includes the following source data, source code, and figure supplement(s) for figure 6:

**Source code 1.** FlyTracker MATLAB script to measure locomotion parameters.

**Source data 1.** Quantitative results used to generate the graphs in *Figure 6A*, *Figure 6—figure supplements 1 and 2*.

**Source data 2.** Quantitative data used to generate graphs in *Figure 6B*.

**Source data 3.** Quantititative data used to generate graph in *Figure 6C*.

**Source data 4.** Quantitative data used to generate graphs in *Figure 6D*.

**Source data 5.** Quantitative data used to generate graph in *Figure 6E*.

**Source data 6.** Quantitative data used to generate graph in *Figure 6F*.

**Source data 7.** Quantitative data used to generate graphs in *Figure 6G*.

**Figure supplement 1.** Locomotion in open arena controls.

**Figure supplement 1—source data 1.** Quantitative data used to generate graph in *Figure 6—figure supplement 1A*.

**Figure supplement 1—source data 2.** Quantitative data used to generate graph in *Figure 6—figure supplement 1B*.

**Figure supplement 2.** Altering *DNT-2* levels induced seizures.

**Figure supplement 2—source data 1.** Quantitative data used to generate graph in *Figure 6—figure supplement 2*, left.

**Figure supplement 2—source data 2.** Quantitative data used to generate graph in *Figure 6—figure supplement 2*, middle.

**Figure supplement 2—source data 3.** Quantitative data used to generate graph in *Figure 6—figure supplement 2*, right.

## Discussion

Our findings indicate that structural plasticity and degeneration in the brain are two manifestations of a shared molecular mechanism that could be modulated by experience. Loss of function for *DNT-2, Toll-6,* and *kek-6* caused cell loss, affected arborisations and synaptogenesis in DANs, and impaired locomotion; neuronal activity increased DNT-2 production and cleavage and remodelled connecting DNT-2 and PAM synapses; and over-expression of *DNT-2* increased TH levels, PAM cell number, dendrite complexity, and synaptogenesis, and it enhanced locomotion and long-term memory.

It was remarkable to find that the number of DANs in the *Drosophila* adult brain is plastic, and this is functionally relevant as it can influence behaviour. We showed that PAM cell number is variables across individuals, that adult-specific gain of *DNT-2* function increases, whereas loss of *DNT-2* or *Toll-6* function decreases, PAM cell number. Loss of *DNT-2* function in mutants, constant loss of *Toll-6* function in DANS and adult-restricted knock-down of either *DNT-2* (in DNT-2 neurons) or *Toll-6* (in Toll-6 neurons and in DANs) all resulted in DAN cell loss, verified with three distinct reporters, and consistently with the increase in DAN apoptosis. Furthermore, DAN cell loss in *DNT-2* mutants could be rescued by the over-expression of *Toll-6* in DANs. Cell loss was also verified using two reporter types (i.e. GAL4-based nuclear reporters and cytoplasmic anti-TH antibodies), multiple GAL4 drivers and mutants, and multiple cell counting methods, including automatic cell counting with DeadEasy plug-ins for His-YFP and nls-Tomato (where the signal was of high contrast and sphericity) and software-assisted manual cell counting for anti-TH (where the signal is more diffuse and less regular in shape). Dead-Easy plug-ins have been used before for reliably counting His-YFP-labelled cells in both larval CNS and adult brains, including KCs (*Kato et al., 2011*; *Forero et al., 2012*; *Losada-Perez et al., 2016*; *Li et al., 2020*; *Harrison et al., 2021*). Thus, the finding that loss of *DNT-2* and *Toll-6* function in the adult brain cause DAN loss is robust. Our findings are reminiscent of the increased apoptosis and cell

loss in adult brains with *Toll-2* loss of function (*Li et al., 2020*), and the support of DAN survival by Toll-1 and Toll-7-driven autophagy (*Zhang et al., 2024*). They are also consistent with a report that loss of function for *DNT-2* or *Toll-6* induced apoptosis in the third-instar larval optic lobes (*McLaughlin et al., 2019*). This did not result in neuronal loss, which was interpreted as due to Toll-6 functions exclusive to glia (*McLaughlin et al., 2019*), but instead of testing the optic lobes, neurons of the larval abdominal ventral nerve cord (VNC) were monitored (*McLaughlin et al., 2019*). In the VNC, Toll-6 and -7 function redundantly and knock-down of both is required to cause neuronal loss in embryos (*McIlroy et al., 2013*), whereas in L3 larvae and pupae the phenotype is compounded by their pro-apoptotic functions (*Foldi et al., 2017*). It is crucial to consider that the DNT-Toll signalling system can have distinct cellular outcomes depending on context, cell type, and time, that is, stage (*Foldi et al., 2017*; *Li et al., 2020*; *Li and Hidalgo, 2021*). Our work shows that in the adult *Drosophila* brain DANs receive secreted growth factors that maintain cellular integrity, and this impacts behaviour. Consistently with our findings, *Drosophila* models of Parkinson's disease reproduce the loss of DANs and locomotion impairment of human patients (*Feany and Bender, 2000*; *Riemensperger et al., 2011*; *Riemensperger et al., 2013*; *Sun et al., 2018*). Dopamine is required for locomotion, associative reward learning, and long-term memory (*Riemensperger et al., 2011*; *Riemensperger et al., 2013*; *Waddell, 2013*; *Sun et al., 2018*; *Boto et al., 2020*; *Adel and Griffith, 2021*). In *Drosophila*, this requires PAM, PPL1, and DAL neurons and their connections to KCs and MBONs (*Heisenberg, 2003*; *Chen et al., 2012*; *Plaçais et al., 2012*; *Aso et al., 2014b*; *Plaçais et al., 2017*; *Boto et al., 2020*; *Adel and Griffith, 2021*; *Huang et al., 2024*). DNT-2 neurons are connected to all these neuron types, which express *Toll-6* and *kek-6,* and modifying their levels affects locomotion and long-term memory. Altogether, our data demonstrate that structural changes caused by altering DNT-2, Toll-6, and Kek-6 modified dopamine-dependent behaviours, providing a direct link between molecules, structural circuit plasticity, and behaviour.

We used neuronal activation as a proxy for experience, but the implication is that experience would similarly drive the structural modification of circuits labelled by neuromodulators. Similar manipulations of activity have previously revealed structural circuit modifications. For example, hyperpolarising olfactory projection neurons increased microglomeruli number, active zone density, and post-synaptic site size in the calyx, whereas inhibition of synaptic vesicle release decreased the number of microglomeruli and active zones (*Kremer et al., 2010*). There is also evidence that experience can modify circuits and behaviour in *Drosophila*. For example, natural exposure to light and dark cycles maintains the structural homeostasis of pre-synaptic sites in photoreceptor neurons, which breaks down in sustained exposure to light (*Sugie et al., 2015*). Prolonged odour exposure causes structural reduction at the antennal lobe and at the output pre-synaptic sites in the calyx, and habituation (*Devaud et al., 2001*; *Pech et al., 2015*). Similarly, prolonged exposure to $CO_2$ caused a reduction in output responses at the lateral horn and habituation (*Sachse et al., 2007*). Our findings are also consistent with previous reports of structural plasticity during learning in *Drosophila*. Hypocaloric food promotes structural plasticity in DANs, causing a reduction specifically in connections between DANs and KCs involved in aversive learning, thus decreasing the memory of the aversive experience (*Çoban et al., 2024*). In contrast, after olfactory conditioning, appetitive long-term memory increased axonal collaterals in projection neurons, and synapse number at KC inputs in the calyx (*Baltruschat et al., 2021*). Our data provide a direct link between a molecular mechanism, synapse formation in a dopaminergic circuit, and behavioural performance. Since behaviour is a source of experience, the discovery that a neurotrophin can function with a Toll and a neuromodulator to sculpt circuits provides a mechanistic basis for how experience can shape the brain throughout life.

Importantly, in humans, structural brain plasticity (e.g. adult neurogenesis, neuronal survival, neurite growth, and synaptogenesis) correlates with anti-depressant treatment, learning, physical exercise, and well-being (*Cotman and Berchtold, 2002*; *Woollett and Maguire, 2011*; *Castrén and Monteggia, 2021*; *Cheng et al., 2023*). Conversely, neurite, synapse, and cell loss correlate with ageing, neuroinflammation, psychiatric, and neurodegenerative conditions (*Holtmaat and Svoboda, 2009*; *Lu et al., 2013*; *Wohleb et al., 2016*; *Forrest et al., 2018*; *Vahid-Ansari and Albert, 2021*; *Wang et al., 2022*). Understanding how experience drives the switch between generative and destructive cellular processes shaping the brain is critical to understand brain function in health and disease. In this context, the mechanism we have discovered could also operate in the human brain. In fact, there is deep evolutionary conservation in DNT-2 vs mammalian NT function (e.g. BDNF), but

some details differ. Like mammalian NTs, full-length DNTs/Spz proteins contain a signal peptide, an unstructured pro-domain, and an evolutionarily conserved cystine knot (CK) of the NT family (**Zhu et al., 2008**; **Arnot et al., 2010**; **Foldi et al., 2017**). Cleavage of the pro-domain releases the mature CK. In mammals, full-length NTs have opposite functions to their cleaved forms (e.g. apoptosis vs cell survival, respectively). However, DNT-2FL is cleaved by intracellular furins, and although DNT-2 can be found both in full-length or cleaved forms in vivo, it is most abundantly found cleaved (**Zhu et al., 2008**; **McIlroy et al., 2013**; **Foldi et al., 2017**). As a result, over-expressed *DNT-2FL* does not induce apoptosis and instead it promotes cell survival (**Foldi et al., 2017**). The same functions are played by over-expressed mature *DNT-2CK* as by *DNT-2FL* (**Zhu et al., 2008**; **Foldi et al., 2017**; **Ulian-Benitez et al., 2017**). In S2 cells, transfected DNT-2CK is not secreted, but when over-expressed in vivo it is functional (**Zhu et al., 2008**; **Foldi et al., 2017**; **Ulian-Benitez et al., 2017**; and this work). In fact, over-expressed *DNT-2CK* also maintains neuronal survival, connectivity, and synaptogenesis (**Zhu et al., 2008**; **Foldi et al., 2017**; **Ulian-Benitez et al., 2017**; and this work). Similarly, over-expressed mature *spz-1-C106* can rescue the *spz-1*-null mutant phenotype (**Hu et al., 2004**) and over-expressed *DNT-1CK* can promote neuronal survival, connectivity, and rescue the *DNT-1* mutant phenotype (**Zhu et al., 2008**). Consistently, both DNT-2FL and DNT-2CK have neuro-protective functions promoting cell survival, neurite growth, and synaptogenesis (**Zhu et al., 2008**; **Sutcliffe et al., 2013**; **Foldi et al., 2017**; **Ulian-Benitez et al., 2017**; and this work). Importantly, over-expressing *DNT-2FL* enables to investigate non-autonomous functions of DNT-2 (**Ulian-Benitez et al., 2017**). We have shown that *DNT-2FL* can induce synaptogenesis non-autonomously in target neurons. Similarly, DNT-2 is a retrograde factor at the larval NMJ, where transcripts are located post-synaptically in the muscle, and DNT-2FL-GFP is taken up from muscle by motoneurons, where it induces synaptogenesis (**Sutcliffe et al., 2013**; **Ulian-Benitez et al., 2017**). Importantly, we have shown that neuronal activity increased production of tagged DNT-2-GFP and its cleavage. In mammals, neuronal activity induces synthesis, release, and cleavage of BDNF, leading to neuronal survival, dendrite growth and branching, synaptogenesis, and synaptic plasticity (i.e. LTP) (**Poo, 2001**; **Horch and Katz, 2002**; **Lu et al., 2005**; **Arikkath, 2012**; **Wang et al., 2022**). Like BDNF, DNT-2 also induced synaptogenesis and increased bouton size.

It may not always be possible to disentangle primary from compensatory phenotypes as Hebbian, homeostatic, and heterosynaptic plasticity can concur (**Forrest et al., 2018**; **Jenks et al., 2021**). Mammalian BDNF increases synapse number, spine size, and LTP, but it can also regulate homeostatic plasticity and LTD, depending on the timing, levels, and site of action (**Poo, 2001**; **Lu et al., 2005**; **Wang et al., 2022**). In this context, neuronal stimulation of DNT-2 neurons induced synapse formation in PAM neuron SMP dendrites, whereas DNT-2 over-expression from DNT-2 neurons increased synapse size at SMP and synapse number in PAM outputs at the mushroom body lobes. These distinct effects could be due to the combination of plasticity mechanisms and range of action. Neuronal activity can induce localised protein synthesis that facilitates local synaptogenesis and stabilises emerging synapses (**Forrest et al., 2018**). In contrast, DNT-2 induced signalling via the nucleus can facilitate synaptogenesis at longer distances in output sites. In any case, synaptic remodelling is the result of concurring forms of activity-dependent plasticity altogether leading to modification in connectivity patterns (**Forrest et al., 2018**; **Jenks et al., 2021**). Long-term memory requires synaptogenesis, and in mammals this depends on BDNF and its role in the protein synthesis-dependent phase of LTP (**Poo, 2001**; **Minichiello, 2009**; **Wang et al., 2022**). BDNF-localised translation, expression, and release are induced to enable long-term memory (**Poo, 2001**; **Lee et al., 2004**; **Minichiello, 2009**; **Wang et al., 2022**). We have shown that similarly over-expressed *DNT-2FL* increased both synaptogenesis in sites involved in reward learning and long-term memory after appetitive conditioning.

The relationship of NTs with dopamine is also conserved. DNT-2 and DAN neurons form bidirectional connectivity that modulates both DNT-2 and dopamine levels. Similarly, mammalian NTs also promote dopamine release and the expression of DA receptors (**Blöchl and Sirrenberg, 1996**; **Guillin et al., 2001**). Furthermore, DAN cell survival is maintained by DNT-2 in *Drosophila*, and similarly DAN cell survival is also maintained by NT signalling in mammals and fish (**Hyman et al., 1991**; **Sahu et al., 2019**). Importantly, we showed that activating DNT-2 neurons increased the levels and cleavage of DNT-2, up-regulated DNT-2 increased *TH* expression, and this initial amplification resulted in the inhibition of cAMP signalling via the dopamine receptor Dop2R in DNT-2 neurons. This negative feedback could drive a homeostatic reset of both DNT-2 and dopamine levels, important for normal brain function. In fact, we showed that alterations in DNT-2 levels could cause seizures. Importantly, alterations

in both NTs and dopamine underlie many psychiatric disorders and neurodegenerative diseases in humans (*Hyman et al., 1991*; *Guillin et al., 2001*; *Berton et al., 2006*; *Forrest et al., 2018*; *Wang et al., 2022*).

We have uncovered a novel mechanism of structural brain plasticity, involving an NT ligand functioning via a Toll and a kinase-less Trk-family receptor in the adult *Drosophila* brain. Toll receptors in the CNS can function via ligand-dependent and ligand-independent mechanisms (*Anthoney et al., 2018*). However, in the context analysed, Toll-6 and Kek-6 function in structural circuit plasticity depends on their ligand DNT-2. This is also consistent with their functions promoting axonal arbour growth, branching, and synaptogenesis at the NMJ (*Ulian-Benitez et al., 2017*). Furthermore, Toll-2 is also neuro-protective in the adult fly brain, and loss of *Toll-2* function caused neurodegeneration and impaired behaviour (*Li et al., 2020*). There are six *spz/DNT*, nine *Toll*, and six *kek* paralogous genes in *Drosophila* (*Tauszig et al., 2000*; *MacLaren et al., 2004*; *Mandai et al., 2009*; *Ulian-Benitez et al., 2017*), and at least seven *Tolls* and three adaptors are expressed in distinct but overlapping patterns in the brain (*Li et al., 2020*). Such combinatorial complexity opens the possibility for a fine-tuned regulation of structural circuit plasticity and homeostasis in the brain.

*Drosophila* and mammalian NTs may have evolved to use different receptor types to elicit equivalent cellular outcomes. In fact, in mammals, NTs function via Trk, p75$^{NTR}$, and Sortilin receptors to activate ERK, PI3K, NFκB, JNK, and CaMKII (*Lu et al., 2005*; *Wang et al., 2022*). Similarly, DNTs with Tolls and Keks also activate ERK, NFκB, JNK, and CaMKII in the *Drosophila* CNS (*McIlroy et al., 2013*; *Foldi et al., 2017*; *Ulian-Benitez et al., 2017*; *Anthoney et al., 2018*). However, alternatively, NTs may also use further receptors in mammals. Trks have many kinase-less isoforms, and understanding of their function is limited (*Fryer et al., 1996*; *Stoilov et al., 2002*). They can function as ligand sinks and dominant negative forms, but they can also function independently of full-length Trks to influence calcium levels, growth cone extension, and dendritic growth and are linked to psychiatric disorders, for example, depression (*Ferrer et al., 1999*; *Yacoubian and Lo, 2000*; *Ohira et al., 2006*; *Carim-Todd et al., 2009*; *Ernst et al., 2009*; *Ohira and Hayashi, 2009*; *Fenner, 2012*; *Tessarollo and Yanpallewar, 2022*). Like Keks, perhaps kinase-less Trks could regulate brain plasticity vs. degeneration.

A functional relationship between NTs and TLRs could exist also in humans, as in cell culture, human BDNF and NGF can induce signalling from a TLR (*Foldi et al., 2017*) and NGF also functions in immunity and neuroinflammation (*Levi-Montalcini et al., 1996*; *Hepburn et al., 2014*). Importantly, TLRs can regulate cell survival, death and proliferation, neurogenesis, neurite growth and collapse, learning and memory (*Okun et al., 2011*). They are linked to neuroinflammation, psychiatric disorders, neurodegenerative diseases, and stroke (*Okun et al., 2011*; *Figueroa-Hall et al., 2020*; *Adhikarla et al., 2021*). Intriguingly, genome-wide association studies have revealed the involvement of TLRs in various brain conditions and potential links between NTs and TLRs in, for example, major depression (*Sharma, 2012*; *Mehta et al., 2018*; *Chan et al., 2020*; *Garrett et al., 2021*). Importantly, alterations in NT function underlie psychiatric, neurological, and neurodegenerative brain diseases (*Lu et al., 2005*; *Martinowich et al., 2007*; *Krishnan and Nestler, 2008*; *Lu et al., 2013*; *Park and Poo, 2013*; *Wohleb et al., 2016*; *Yang et al., 2020*; *Casarotto et al., 2021*; *Wang et al., 2022*), and BDNF underlies the plasticity inducing function of anti-depressants (*Lu et al., 2013*; *Casarotto et al., 2021*; *Wang et al., 2022*). It is compelling to find out whether and how these important protein families – NTs, TLRs, and kinase-less Trks – interact in the human brain.

## Conclusion

We provide a direct link between structural circuit plasticity and behavioural performance by a novel molecular mechanism. The neurotrophin DNT-2 and its receptors Toll-6 and the kinase-less Trk family Kek-6 are linked to a dopaminergic circuit. Neuronal activity boosts DNT-2, and DNT-2 and dopamine regulate each other homeostatically. Dopamine labels the circuits engaged and DNT-2, a growth factor, with its receptors Toll-6 and Kek-6, drives structural plasticity in these circuits, enhancing dopamine-dependent behavioural performance. These findings mean that DNT-2 is a plasticity factor in the *Drosophila* brain that could enable experience-dependent behavioural enhancement. Whether NTs can similarly function with TLRs and kinase-less Trks remains to be explored. As behaviour is a source of experience, this has profound implications for understanding brain function and health.

# Materials and methods

## Key resources table

| Reagent type (species) or resource | Designation | Source or reference | Identifiers | Additional information |
|---|---|---|---|---|
| Genetic reagent (*Drosophila melanogaster*) | DNT2 Gal4 [CRISPR] | This study | Hidalgo Lab | See Materials and Methods, Molecular Biology and *Figure 1*. |
| Genetic reagent (*D. melanogaster*) | Toll-6 Gal4 [MIO2127] | Hidalgo Lab; *Li et al., 2020* | Hidalgo Lab | |
| Genetic reagent (*D. melanogaster*) | kek-6 Gal4 [MI13953] | Hidalgo Lab; *Ulian-Benitez et al., 2017* | Hidalgo Lab | |
| Genetic reagent (*D. melanogaster*) | Toll-2 Gal4 [pTV] | Hidalgo Lab; *Li et al., 2020* | Hidalgo Lab | |
| Genetic reagent (*D. melanogaster*) | Toll-7 Gal4 [MI13963] | Hidalgo Lab; *Li et al., 2020* | Hidalgo Lab | |
| Genetic reagent (*D. melanogaster*) | Toll-5 Gal4 [CRISPR] | Hidalgo Lab; *Li et al., 2020* | Hidalgo Lab | |
| Genetic reagent (*D. melanogaster*) | Toll-8 Gal4 [MD806] | Bloomington *Drosophila* Stock Center (BDSC) | RRID:BDSC_36548 | |
| Genetic reagent (*D. melanogaster*) | MB320C-Gal4 (PPL1-ped) [1w[1118]; P{y[+t7.7] w[+mC]=R22B12-GAL4.DBD} attP2 PBac{y[+mDint2] w[+mC]=ple-p65.AD} VK00027118]; P{y[+t7.7] w[+mC]=R71D01-p65.AD} attP40; P{y[+t7.7] w[+mC]=R58F02-GAL4.DBD} attP2 | BDSC | RRID:BDSC_68253 | |
| Genetic reagent (*D. melanogaster*) | MB301B-Gal4 (PAM-b2b'2) w[1118]; P{y[+t7.7] w[+mC]=R71D01-p65.AD} attP40; P{y[+t7.7] w[+mC]=R58F02-GAL4.DBD} attP2 | BDSC | RRID:BDSC_68311 | |
| Genetic reagent (*D. melanogaster*) | TH-LexA(II) | Gift from Davis Ronald (via Serge Birman) | | |
| Genetic reagent (*D. melanogaster*) | TH-Gal4,R58E02-Gal4 | Gift from Serge Birman | | |
| Genetic reagent (*D. melanogaster*) | R58E02-Gal4 | BDSC | RRID:BDSC_41347 | |
| Genetic reagent (*D. melanogaster*) | R58E02-LexA | BDSC | RRID:BDSC_52740 | |
| Genetic reagent (*D. melanogaster*) | R14C08-LexA | BDSC | RRID:BDSC_52473 | |
| Genetic reagent (*D. melanogaster*) | Dop1R1-LexA | Gift from *Deng et al., 2019* | | |
| Genetic reagent (*D. melanogaster*) | Dop2R-LexA | Gift from *Deng et al., 2019* | | |
| Genetic reagent (*D. melanogaster*) | Gad-LexA | Gift from Yi Rao | | |
| Genetic reagent (*D. melanogaster*) | VT49239-LexA (DAL-LexA) | Gift from Ann-Shyn Chiang | | |
| Genetic reagent (*D. melanogaster*) | Tdc2-LexA | Gift from Carolina Rezaval | | |

*Continued on next page*

*Continued*

| Reagent type (species) or resource | Designation | Source or reference | Identifiers | Additional information |
|---|---|---|---|---|
| Genetic reagent (*D. melanogaster*) | Canton-S | Gift from Kei Ito | | |
| Genetic reagent (*D. melanogaster*) | Oregon | Hidalgo Lab | | |
| Genetic reagent (*D. melanogaster*) | MCFO: w[1118] P{y[+t7.7] w[+mC]=hs-FLPG5.PEST}attP3; P{y[+t7.7] w[+mC]=10xUAS(FRT.stop)myr::smGdP-OLLAS}attP2 PBac{y[+mDint2] w[+mC]=10xUAS(FRT.stop)myr::smGdP-HA} VK00005 P{10xUAS(FRT.stop)myr::smGdP-V5-THS-10xUAS(FRT.stop)myr::smGdP-FLAG}su(Hw)attP1 | BDSC | RRID:BDSC_64086 | |
| Genetic reagent (*D. melanogaster*) | UAS-DNT2FL-GFP | Hidalgo Lab; *Ulian-Benitez et al., 2017* | Hidalgo Lab | |
| Genetic reagent (*D. melanogaster*) | UAS-FlyBow1.1 (2609) | Gift from Iris Salecker (*Hadjieconomou et al., 2011*) | | |
| Genetic reagent (*D. melanogaster*) | hs Flp;;UAS-FlyBow2.0 | Gift from Iris Salecker (*Ferrer et al., 1999*; *Hadjieconomou et al., 2011*) | | |
| Genetic reagent (*D. melanogaster*) | 13xLexAop-nls-tdTomato | Gift from B. Pfeiffer | | |
| Genetic reagent (*D. melanogaster*) | UAS-DNT-2RNAi [VDRC49195] | VDRC | VDRC49195 | |
| Genetic reagent (*D. melanogaster*) | UAS-Toll6RNAi [P{GD35}v928] | VDRC | VDRC:928 FBst0471444 | |
| Genetic reagent (*D. melanogaster*) | UAS-Toll-6[CY] | Hidalgo Lab; *McIlroy et al., 2013* | Hidalgo Lab | |
| Genetic reagent (*D. melanogaster*) | UAS-KeK6RNAi [KK-109681] | VDRC | VDRC:109681 FBgn0039862 | |
| Genetic reagent (*D. melanogaster*) | TransTango: UAS-myrGFP.QUAS-mtdTomato-3xHA (attp8); transTango (attp40) | BDSC | RRID:BDSC_77124 | |
| Genetic reagent (*D. melanogaster*) | 13LexAop-CD8::GFP, 10UAS-CD8::RFP | BDSC | RRID:BDSC_32229 | |
| Genetic reagent (*D. melanogaster*) | UAS-FB1.1 (406) | Gift from Iris Salecker (*Hadjieconomou et al., 2011*) | | |
| Genetic reagent (*D. melanogaster*) | UAS-DenMarkRFP, UAS-Dsyd1GFP | Gift from Carolina Rezaval | | |
| Genetic reagent (*D. melanogaster*) | MB247-Gal80 | BDSC | RRID:BDSC_64306 | |
| Genetic reagent (*D. melanogaster*) | DNT2[37]/TM6BlacZ | Hidalgo Lab; *Ulian-Benitez et al., 2017* | | |
| Genetic reagent (*D. melanogaster*) | DNT2[18] | Hidalgo Lab; *Ulian-Benitez et al., 2017* | | |
| Genetic reagent (*D. melanogaster*) | tubGal80[ts] on 2nd | BDSC | Hidalgo Lab | |

*Continued on next page*

*Continued*

| Reagent type (species) or resource | Designation | Source or reference | Identifiers | Additional information |
|---|---|---|---|---|
| Genetic reagent (*D. melanogaster*) | Toll-6[31]/TM6BlacZ | Hidalgo Lab (*McIlroy et al., 2013*) | | |
| Genetic reagent (*D. melanogaster*) | UAS-dop2R-RNAi [TRiP.HMC02988}attP40] | BDSC | RRID:BDSC_50621 | |
| Genetic reagent (*D. melanogaster*) | UAS-Toll6-RNAi [TRiP.HMS04251}attP2] | BDSC | RRID:BDSC_56048 | |
| Genetic reagent (*D. melanogaster*) | UAS-Brp-s-mCherry (II) | Gift from Stephan Sigrist | | |
| Genetic reagent (*D. melanogaster*) | UAS-Brp-s-mCherry (III) | Gift from Stephan Sigrist | | |
| Genetic reagent (*D. melanogaster*) | UAS-CD8::GFP (II) | Hidalgo Lab | | |
| Genetic reagent (*D. melanogaster*) | BacTrace 806: w; LexAop2-Syb::GFP-P10(VK37) LexAop-QF2::SNAP25::HIVNES::Syntaxin(VK18) / CyO; UAS-B3Recombinase(attP2) UAS<B3Stop <BoNT/A(VK5) | *Cachero et al., 2020* | | |
| Genetic reagent (*D. melanogaster*) | UAS-UAS-Epac1-camps-50A | BDSC | RRID:BDSC_25408 | |
| Genetic reagent (*D. melanogaster*) | 20x UAS V5 Syn Cs Chrimson td tomato | Gift from B. Pfeiffer | | |
| Genetic reagent (*D. melanogaster*) | 13XLexAop2-IVS-GCaMP6s-p10 su(Hw)attP1 | BDSC | RRID:BDSC_44274 | |
| Genetic reagent (*D. melanogaster*) | 13XLexAop2-IVS-GCaMP6s-p10 su(Hw)attP5 | BDSC | RRID:BDSC_44590 | |
| Genetic reagent (*D. melanogaster*) | P{w[+mC]=UAS-syt.eGFP}2 | BDSC | RRID:BDSC_6925 | |
| Genetic reagent (*D. melanogaster*) | UAS-homer-GCaMP3 on 2nd | Gift from André Fialá | | |
| Antibody | Mouse monoclonal anti-brp (nc82) | Developmental Studies Hybridoma Bank (DSHB), IA | Cat# nc82; RRID:AB_2314866 | 1:10 |
| Antibody | Rabbit polyclonal anti-GFP | Thermo Fisher | Cat# A-11122; RRID:AB_221569 | 1:250 |
| Antibody | Mouse monoclonal anti-GFP | Thermo Fisher | Cat# A-11120; RRID:AB_221568 | 1:250 |
| Antibody | Chicken polyclonal anti-GFP | Aves | Cat# GFP1010; RRID:AB_2307313 | 1:500 |
| Antibody | Rabbit polyclonal anti-FLAG | Sigma | Cat# F7425 | 1:50 |
| Antibody | Mouse monoclonal anti-V5 | Invitrogen | Cat# # R960-25; RRID:AB_2556564 | 1:50 |
| Antibody | Chicken polyclonal anti-HA | Aves | Cat# ET-HA100; RRID:AB_2313511 | 1:50 |
| Antibody | Rabbit polyclonal anti-Vglut | Gift from Hermann | | 1:500 |
| Antibody | Mouse monoclonal anti-TH | Immunostar | Cat# 22941; RRID:AB_572268 | 1:250 |
| Antibody | Rabbit polyclonal anti-TH | Novus Biologicals | Cat# NB300-109 | 1:250 |
| Antibody | Rabbit polyclonal anti-DsRed | Clontek | Cat# 632496; RRID:AB_10013483 | 1:250 |
| Antibody | Mouse monoclonal anti-ChAT4B1 | DHSB | Cat# ChAT4B1; RRID:AB_528122 | 1:250 |
| Antibody | Rabbit polyclonal anti-5-HT | Immunostar | Cat#20080; RRID:AB_572263 | 1:500 |

*Continued on next page*

*Continued*

| Reagent type (species) or resource | Designation | Source or reference | Identifiers | Additional information |
|---|---|---|---|---|
| Antibody | Rabbit polyclonal anti-DCP-1 | Cell Signalling | Cat# 8578S | 1:250 |
| Antibody | Alexa Flour 488 goat anti-mouse | Thermo Fisher | Cat# A-11001; RRID:AB_2534069 | 1:500 |
| Antibody | Alexa Flour 488 donkey anti-rabbit | Thermo Fisher | Cat# A-21206; RRID:AB_2435792 | 1:500 |
| Antibody | Alexa Flour 488 goat anti-rabbit (Fab')2 | Thermo Fisher | Cat# A-11070; RRID:AB_2534114 | 1:500 |
| Antibody | Alexa Flour 488 goat anti-chicken | Thermo Fisher | Cat# A-11039; RRID:AB_2534096 | 1:500 |
| Antibody | Alexa Four 546 goat anti-rabbit | Thermo Fisher | Cat# A-11035; RRID:AB_2534093 | 1:500 |
| Antibody | Alexa Four 546 goat anti-mouse | Thermo Fisher | Cat# A-11003; RRID:AB_25334071 | 1:500 |
| Antibody | Alex Four 647 goat anti-rabbit | Thermo Fisher | Cat# A-21245; RRID:AB_2535813 | 1:500 |
| Antibody | Alex Four 647 goat anti-mouse | Thermo Fisher | Cat# A-21236; RRID:AB_2535905 | :500 |
| Chemical compound, drug. | TRIzol | Ambion | Cat# AM9738 | |
| Commercial assay or kit | DNA-*free* DNA Removal Kit | Thermo Fisher | Cat# AM1906 | |
| Commercial assay or kit | GoScript Reverse Transcriptase | Promega | Cat# 237815 | |
| Chemical compound, drug | SensiFAST synGreen Mix | Bioline | Cat# B2092020 | |
| Chemical compound, drug | PFA | Sigma-Aldrich | Car# P6148 | |
| Chemical compound, drug | Triton X-100 | | | |
| Chemical compound, drug | Normal goat serum | Vector Laboratories | Cat# S1000 | |
| Chemical compound, drug | 3-Octanol | Sigma-Aldrich | Cat# 218405 | |
| Chemical compound, drug | 4-Methylcyclohexaniol | Sigma-Aldrich | Cat# 153095 | |
| Chemical compound, drug | Mineral oil | Sigma-Aldrich | Cat# 330760 | |
| Commercial assay or kit | qPCR plate | GeneFlow | Cat# P3-0292 | |
| Software | Prism6 | GraphPad, CA | RRID:SCR_002798 | |
| Software | Fiji/ImageJ | Fiji | RRID:SCR_002285 | |
| Software | DeadEasy-Central Brain | *Figure 3—source code 1*. ImageJ plug-in originally published in *Li et al., 2020* | DeadEasy_Central_Brain in UBIRA; https://edata.bham.ac.uk/ | |
| Software | DeadEasy-Dopaminergic neuron | *Figure 3—source code 2*. ImageJ plug-in | DeadEasy_Dopaminergic3DNew ni UBIRA; https://edata.bham.ac.uk/ | |
| Software | Imaris | Bitplane | RRID:SCR_007370 | |
| Software | Adobe Illustrator | Adobe CS | RRID:SCR_010279 | |
| Software | Adobe Photoshop | Adobe CS | RRID:SCR_014199 | |

Continued

| Reagent type (species) or resource | Designation | Source or reference | Identifiers | Additional information |
|---|---|---|---|---|
| Software; algorithm | FlyTracker | Eyrun Eyjolfsdottir, Steve Branson, Xavier P. Burgos-Artizzu, Eric D. Hoopfer, Jonathan Schor, David J Anderson, Pietro Perona. *Eyjolfsdottir et al., 2014*, Computer Vision – ECCV 772–787 | https://kristinbranson.github.io/FlyTracker/ | |
| Software | MATLAB | MathWorks | RRID:SCR_001622 | |
| Other | Scope Fly Cell Atlas | *Davie et al., 2018*; *Davie et al., 2018* | https://scope.aertslab.org/#/FlyCellAtlas/*/welcome | |

## Genetics

Mutants: *DNT2³⁷* and *DNT2¹⁸* are protein null (*Foldi et al., 2017*; *Ulian-Benitez et al., 2017*). *Toll6³¹* is a null mutant allele (*McIlroy et al., 2013*). Driver lines: *DNT2-Gal4* is a CRISPR/Cas9-knock-in allele, with GAL4 at the start of the gene (this work, see below). *Toll-6-Gal4* was generated by RMCE from *MIMIC Toll-6ᴹᴵᴼ²¹²⁷*; *kek6-Gal4* from *MIMIC Kek6ᴹᴵ¹²⁹⁵³* (*Ulian-Benitez et al., 2017*; *Li et al., 2020*). *MB320C-Gal4 (BSC68253), MB301B-Gal4 (BSC68311), R58E02-Gal4 (BSC41347), MB247-Gal80 (BSC64306), R58E02-LexA (BSC52740), R14C08-LexA (BSC52473)* were from the Bloomington Drosophila Stock Center (BDSC). *Dop1R2-LexA, Dop2R-LexA, Gad-LexA* were kindly provided by Yi Rao; *TH-LexA* (gift from Ron Davis), *TH-Gal4,R58E02-GAL4* (gift from Serge Birman); *G0431-Gal4 (DAL-GAL4), VT49239 nls LexA (DAL LexA)* (gifts from Ann-Shyn Chiang); tubGal80ts, Tdc-LexA. Reporter lines: *UAS-CD8::GFP,* for membrane tethered GFP; *UAS-histone-YFP,* for YFP-tagged nuclear histone; *UASFlybow1.1,* constant membrane tethered expression (gift from Iris Salecker); *13xLexOp-nls-tdTomato,* nuclear Tomato (gift from B. Pfeiffer); *UASCD8::RFP, LexAopCD8::GFP (BSC32229),* for dual binary expression; *UAS-homer-GCaMP* for PSDs (gift from André Fialá), *UAS-syt.eGFP (BSC6925)* and *LexAop-Syt-GCaMP (BSC64413)* for pre-synaptic sites. For connectivity: *UAS-DenMarkRFP, UAS-Dsyd1GFP* (gift from Carolina Rezaval); TransTango: *yw UAS-myrGFP,QUAS-mtdTomato-3xHA attP8; Trans-Tango@attP40 (BSC77124);* BAcTrace 806 (*w;LexAop2-Syb::GFP-P10(VK37) LexAop-QF2::SNAP25::HIVNES::Syntaxin(VK18)/CyO; UAS-B3Recombinase (attP2) UAS<B3Stop<BoNT/A (VK5) UAS<B3Stop<BoNT/A(VK27) QUAS-mtdTomato::HA/TM2*): MCFO clones: *hs-FLPG5.PEST;; 10xUAS (FRT.stop) myr::smGdP-OLLAS 10xUAS (FRT.stop) myr::smGdP-HA 10xUAS (FRT.stop) myr::smGdP-V5-THS-10xUAS (FRT.stop) myr::smGdP-FLAG (BSC64086).* Optogenetic activation: *20x UAS-V5-Syn-CsChrimson td tomato* (gift from B. Pfeiffer). For thermogenetic activation: *UASTrpA1@attP2 (BSC26264)* and *UAS-TrpA1@attP216 (BSC26263)*. UAS gene over-expression: *UAS-DNT2CK, UAS-DNT2FL-GFP, UAS-DNT2-FL-47C, UAS-Toll-6ᶜʸ* (*McIlroy et al., 2013*; *Foldi et al., 2017*). UAS-RNAi knock-down: *UAS-DNT2RNAi (VDRC49195), UAS-Toll6RNAi (VDRC928), y¹v¹; UAS-Toll6RNAi[Trip.HMS04251] (BSC 56048), UAS-kek6RNAi (VDCR 109681), y¹v¹; UAS-Dop2R-RNAi[TRiP.HMC02988](BSC50621).*

## Molecular biology

*DNT-2GAL4* was generated by CRISPR/Cas9 enhanced homologous recombination. 1 kb long 5′ and 3′homology arms (HA) were amplified by PCR from genomic DNA of wild-type flies using primers for 5′HA: atcgcaccggtttttacaggcacccatgtctga containing AgeI cutting site and cttgacgcggccgcTGTCAATTCATTCGCCGTCGAT containing NotI cutting site. 3′ HA primers were tattaggcgcgccATGACAAAAAGTATTAAACGTCCGCCC containing AscI cutting site and tactcgactagtgaagcacacccaaaataccc agg containing SpeI cutting site. HAs were sequentially cloned by conventional cloning into pGEM-T2AGal4 vector (Addgene #62894). For gRNA cloning, two 20-nucleotide gRNA oligos (gtcGACAAGTTCTTCTTACCTATG and aaacCATAGGTAAGAAGAACTTGTC) were designed using Optimal Target

Finder. BbsI enzyme sites were added: gtc(g) at the 5′ end of sense oligo, and aaac at the 5′ end of the antisense oligo. gRNA is located at 41 bp downstream of the start codon of DNT-2 within the first coding exon. The gRNA was cloned into pU6.3 using conventional ligation. The two constructs were injected in Cas9 bearing flies, and red fluorescent (3xP3-RFP) transformants were selected and balanced, after which 3xP3-RFP was removed with CRE-recombinase.

## qRT-PCR

qRT-PCR was carried out from 20 whole-adult fruit fly heads, frozen in liquid nitrogen before homogenising in 50 ml Trizol (Ambion #AM9738), followed by a stand RNA extraction protocol. RNA was treated with DNase treatment (Thermo Fisher # AM1906) to remove genomic DNA. 200 ng of RNA was used for cDNA synthesis following GoScript Reverse Transcriptase (Promega #237815) protocol. Sample was diluted 1:4 with nuclease-free $H_2O$. Standard qPCR was performed with SensiFast syb Green Mix (Bioline #B2092020) in ABI qPCR plate (GeneFlow #P3-0292) and machine. To amplify TH mRNA, the following primers were used: TH-F: CGAGGACGAGATTTTGTTGGC and TH-R: TTGA GGCGGACCACCAAAG. GAPDH was used as a housekeeping control. Reactions were performed in triplicate. Specificity and size of amplification products were assessed using melting curve analyses. Target gene expression relative to reference gene is expressed as a value of 2-ΔΔCt (where Ct is the crossing threshold).

## Conditional expression

Multi-Colour Flip-Out clones: *DNT2-Gal4, Toll6-Gal4,* and *kek6-Gal4* were crossed with *hsFLP::PEST;;MCFO* flies, and female offspring were collected and heat-shocked at 37°C in a water bath for 15 min, then kept for 48 hr at 25°C before dissecting their brains. *TransTango:* DNT2-Gal4 or Oregon female virgins were crossed with *TransTango* males, progeny flies were raised at 18°C constantly and 15 days after eclosion, and female flies were selected for immunostaining. Thermogenetic activation with *TrpA1:* Fruit flies were bred at 18°C from egg laying to 4 days post-adult eclosion, then shifted to 29°C in a water bath for 24 hr followed by 24 hr recovery at room temperature for over-expressed *DNT-2FL-GFP*; for the other experiments, after breeding as above, adult flies were transferred to an incubator at 30°C, kept there for 24 hr, and then brains were dissected. Conditional gene over-expression and RNAi knock-down: Flies bearing the temperature-sensitive GAL4 repressor tubGal80[ts] were kept at 18°C from egg laying to adult eclosion, then transferred to 30°C incubator for 48 hr for Dcp-1+ and cell counting experiments and for 120 hr for TH+ cell counting.

## Immunostainings

Adult fruit fly female brains were dissected (in PBS), fixed (in 4% paraformaldehyde, room temperature, 20–30 min), and stained following standard protocols. Primary antibodies and their dilutions were as follows: mouse anti-Brp (nc82) 1:10 (DSHB); rabbit anti-GFP 1:250 (Thermo Fisher); mouse anti-GFP 1:250 (Thermo Fisher); chicken anti-GFP 1:500 (Aves); rabbit anti-FLAG 1:50 (Sigma); mouse anti-V5 1:50 (Invitrogen); chicken anti-HA 1:50 (Aves); rabbit anti-VGlut 1:500 (gift from Hermann); mouse anti-TH 1:250 (Immunostar); rabbit anti-TH 1:250 (Novus Biologicals); rabbit anti-DsRed 1:250 (Clontek); mouse anti-ChAT4B1 1:250 (DSHB); rabbit anti-5-HT 1:500 (Immunostar); and rabbit anti-DCP-1 1:250 (Cell Signalling). Seconday antibodies were all used at 1:500 and all were from Thermo Fisher: Alexa Flour 488 goat anti-mouse, Alexa Flour 488 donkey anti-rabbit, Alexa Flour 488 goat anti-rabbit (Fab')2, Alexa Flour 488 goat anti-chicken, Alexa Four 546 goat anti-rabbit, Alexa Four 546 goat anti-mouse, Alex Four 647 goat anti-rabbit, and Alex Four 647 goat anti-mouse.

## Microscopy and imaging

### Laser scanning confocal microscopy

Stacks of microscopy images were acquired using laser scanning confocal microscopy with either Zeiss LSM710, 900, or Leica SP8. Brains were scanned with a resolution of 1024 × 1024, with Leica SP8 ×20 oil objective and 1 mm step for whole brain and DCP-1 stainings, ×40 oil objective and 1 mm step for central brain. Resolution of 1024 × 512 was used for analysing PAM clusters with 0.96 mm step for cell counting; 0.5 mm step for neuronal morphology; and ×63 oil objective with 0.5 mm step for neuronal connections. Acquisition speed in Leica SP8 was 400 Hz, with no line averaging. Resolution of 3072 × 3072 was used for single-image analysis of synapses using either Leica SP8 or Zeiss LSM900 and

Airyscan acquisition with ×40 water objective speed 6, and average 4, or with 1024 × 512, with ×40 oil lens 2× zoom and 0.35 mm step. TH counting in PAM were scanned with Zeiss 710 with a resolution 1024 × 1024, ×40 oil objective, step 1 mm, speed 8. Zeiss LSM900 Airyscan with a resolution of 1024 × 1024, ×40 water objective, speed 7, 0.7 zoom, and 0.31 µm step size was used for acquisition of optical sections of synapses in PAM neurons.

## Optogenetics and Epac1 FRET two-photon imaging

To test whether DNT-2 neurons can respond to dopamine via the Dop2R inhibitory receptor, we used the cAMP sensor Epac1 and two-photon confocal microscopy. Epac1 is FRET probe, whereby data are acquired from CFP and YFP emission and lower YFP/CFP ratio reveals higher cAMP levels. DNT-2Gal4 flies were crossed to *UAS-CsChrimson UAS Epac1* flies to stimulate DNT-2 neurons and detect cAMP levels in DNT2 neurons. 1–3-day-old *DNT2Gal4>UASCsChrimson, UAS Epac1* flies were collected and separated in two groups. Flies bearing *DNT2Gal4 UASCsChrimson UAS Epac1 UASDop2RRNAi* were fed on 50 µM all-trans retinal food for at least 3 days prior to imaging and kept in constant darkness prior to the experiment.

Optogenetic stimulation of fly brains expressing CsChrimson in DNT-2 neurons was carried out using a sapphire 588 nm laser in a two-photon confocal microscope. For acquisition of YFP and CFP data from Epac1 samples, an FV30-FYC filter was applied using a 925 nm laser for both YFP and CFP imaging. The stimulation laser was targeted onto DNT-2 neuron projections in the SMP region for 20 s. Acquisition region of interest (ROI) was at DNT-2A cell bodies with a frame rate of around 10 Hz. The first acquisition started 10 s before the 20 s stimulation, and consequential acquisition was done every 30 s for 10 cycles.

Image analysis of Epac data was carried out using ImageJ. The two channels (YFP and CFP) were separated, and the ratio of YFP/CFP for each pixel was calculated using the ImageJ>Image Calculator by diving YFP channel by CFP channel. The obtained result of YFP/CFP ratio was saved, the mean ratio of YFP/CFP in the ROI was calculated for each time point, and 11 time points were used. The 11 values represent the ratio of YFP/CFP change in the cell body upon stimulation, with 30 s interval and repeated 10 times.

## Cell counting

To count cells labelled with nuclear reporters (e.g. Histone-YFP, nls-tdTomato) and Dcp-1+ cells, where signal is of high intensity, contrast, and sphericity, we adapted the DeadEasy Central Brain ImageJ plug-in (*Li et al., 2020*) for automatic cell counting in adult brains (*Figure 3—source code 1*). Dead-Easy plug-ins automatically identify and count cells labelled with nuclear reporters in 3D stacks of confocal image in the nervous system of embryos (*Forero et al., 2010a*; *Forero et al., 2010b*), larvae (*Kato et al., 2011*; *Forero et al., 2012*; *Losada-Perez et al., 2016*; *Harrison et al., 2021*), and adult (*Li et al., 2020*) *Drosophila*. DeadEasy plug-ins are accurate at counting cells sparsely labelled with nuclear markers, and importantly, treat all genotypes objectively and equally yielding reliable data. Here, adult brains expressing Histone-YFP or nls-td-tomato reporters were dissected, fixed, and scanned without staining them. DeadEasy Central Brain was used with threshold set to 75.

To count the TH-labelled PAMs, where both the signal and the labelled cell shape are more irregular, we used assisted manual cell counting using two methods. First, we developed a plug-in called DeadEasy Dopaminergic3D (*Figure 3—source code 2*) as follows. A median filter was used to reduce Poison noise, without having large losses at the edges. Then, a 3D morphological closing was performed. Next, all very dark pixels were assigned a value of zero. To mark each cell, each chasm in the image was found using a 3D extended h-minimal transform. As more than one local minimum can be found within each cell, which would result in counting a cell multiple times, a 3D inverse dome detection was performed, and then labelled. Thus, each inverse dome was used as a seed to identify each cell. Once the seeds were obtained, a 3D watershed transformation was performed to recover the shape of the cells. Then, we ran DeadEasy DAN on our raw data to obtain a results stack of images and formed a merged stack between the raw and result stacks to manually add any missing cells. This assisted cell counting method was effective at producing accurate cell counts with less labour and time than conventional manual counting and worked well for some genotypes (*Figure 3D*). However, it was less effective with RNAi knock-down genotypes, where the signal can be less intense, for which

TH+ cells were either corrected manually or counted manually, assisted by the ImageJ cell counter instead.

### Dendrite analysis with Imaris

To analyse dendritic complexity, image data were processed with Imaris using the 'Filaments' module with the default algorithm and 'Autopatch function'. A simple ROI with parallelepiped shape was delimited. Thresholds were set for the largest and the smallest diameter of the dendrite, and this was consistent across samples within the same experiment. The starting point threshold was adjusted to only represent the soma of the neurons, and the 'Seed Points threshold' to match the branches of the neurons. 'Remove Seed Points Around Starting Points' and 'Remove Disconnected Segments' were chosen, keeping the default values. The threshold for background substation and local contrast was consistent across all samples within an experiment. The 'Edit function' within the Filaments module was used to correct any inaccuracy detected in the resulting tracing. The number of dendritic branches, dendritic segments, dendritic branch points, and dendritic terminal points were collected to compare the differences between the groups.

### Vesicles, synapses, and PSD analysis with Imaris

To analyse the number and volume of Homer-GCaMP GFP+ PSDs, Syt-GCaMP GFP+ pre-synaptic sites and DNT-2FLGFP+ vesicles, optical section images of confocal stacks through the brain were processed with the Imaris 'Spot function'. To analyse the number and volume of Homer-GCaMP GFP+ PSDs, the 'Surface module' from Imaris was used to restrict an ROI. Then, 'Absolute Intensity Thresholding' method was applied to each sample choosing the same cutoff each time. The resulting surface was applied to mask the original scan. The masked image was processed using 'Image Processing module' from Imaris. Background subtraction followed by threshold cutoff filters were applied. Afterwards, the 'Spots module' was used as explained below.

An ROI was determined for Syt-GCaMP GFP+ pre-synaptic sites and DNT-2FLGFP+ vesicles using the 'Surface module' with the 'Edit Manually' option 'Algorithm'. The ROI for the SMP region started in the slide immediately after the last slide where the soma of PAM neurons was detectable and finished in the last one where the dendrite was visible. The SMP region laterally was delimited by the black space given by the α lobe position. For the MB lobe region, the ROI started in the first slide where γ5,β'2 and β2 was visible (*Aso et al., 2014a*) and finished in the last slide where this structure was appreciable. The surface was used to create a 'Masked Channel', which a posteriori was used to determine the spots using the Spots module.

The 'Spots module' algorithm was set to 'Different Spot Sizes'. An Estimated XY Diameter was set according to each experiment group using the same within an experiment. 'Background Subtraction' option was selected. 'Intensity Center Filter' was used. 'Spot Region' type was determined from 'Local Contrast', and the 'Region Threshold' according to the 'Region Border'. Setting of the threshold was consistent across genotypes.

## Behaviour

### Startle-induced negative geotaxis assay

Startle-induced negative geotaxis assay was carried out as described in *Sun et al., 2018*. Groups of approximately 10 male flies of the same genotype were placed in a fresh tube one night before the test, after which flies were transferred to a column formed with two empty tubes 15 cm long and 2 cm and then habituated for 30 min. Columns were tapped 3–4 times, flies fell to the bottom, and then climbed upwards. Multiple rounds of testing were performed 3–7 times in a row per column. The process was filmed and films were analysed. Flies were scored during the first 15 s after the tapping, and those that climbed above 13 cm and those that climbed below 2 cm were counted separately. Results given are mean ± SEM of the scores obtained with 10 groups of flies per genotype. The performance index (PI) is defined as $\frac{1}{2}[(ntot + ntop - nbot)/ntot]$, where ntot, ntop, and nbot are the total number of flies, the number of flies at the top, and the number of flies at the bottom, respectively. The assay was carried out at 25°C, 55% humidity. Flies with tubGal80[ts] to conditionally overexpress or knock-down were shifted from 18 to 30° at eclosion and kept for 5 days at 30°C to induce Gal4. Experiments were carried in an environmental chamber at 31°C, 60% humidity or in a humidity and temperature-controlled behaviour lab always kept at 25°C.

## Spontaneous locomotion in an open arena

Male flies of each genotype were collected and kept in groups of 10–20 flies in vials containing fresh fly food for 5–9 days. Before filming, three male flies from one genotype were transferred into a 24 mm well of a multi-well plate using an aspirator and habituated for 15–20 min. The multi-well plate with transparent lid and bottom was placed on a white LED light pad (XIAOSTAR Light Box) and either inside a light-shielding black box (PULUZ, 40 * 40 * 40 cm) in a room with constant temperature (25°C) and humidity (55%) to maintain stable environmental conditions (*Figure 6D*) or inside a temperature-controlled environmental chamber at 18°C or 30°C (*Figure 6E and F*). The locomotion behaviour of freely moving flies was filmed with a camera (Panasonic, HC-V260) in the morning from ZT1-ZT4 and for 10 min at a frame rate of 25 fps. The 10 min videos were trimmed (from 00:02:00 to 00:07:00) to 5 min videos for analysis using FlyTracker software (*Eyjolfsdottir et al., 2014*) for *Figure 6E and F* and this software, with a slight modification for *Figure 6D* (*Figure 6—source data 1*). For the *DNT-2* mutants and over-expression of Toll-6$^{CY}$ experiments, flies were bred and tested at 25°C. To test over-expression of *DNT-2* with *tubGAL80$^{ts}$; DNT2-Gal4>UAS-DNT-2FL*, flies were raised at 18°C until eclosion, and controls were kept and tested at 18°C; test groups were transferred directly after eclosion to 30°C for 5 days and tested in an environmental chamber kept at 30°C, 60%. For thermo-genetic activation of DNT2 neurons using TrpA1 (DNT-2GAL4>UASTrpA1), flies were bred at 18°C and kept at 18°C for 7–9 days post-eclosion. Following habituation at 18°C for 20 min in the multi-well plates, they were transferred to the 30°C chamber 10 min before filming to activate TrpA1 and then filmed for the following 10 min. Fly locomotion activity was tracked using FlyTracker (https://kristinbranson.github.io/FlyTracker/index.html) and calculated (distance and speed) in MATLAB (*Eyjolfsdottir et al., 2014*) using the raw data generated from the tracking procedure (see also *Figure 6—source code 1* for *Figure 6D*). The 'walking distance' was calculated as the sum of the distance flies moved, and the 'walking speed' was the speed of flies only when they were walking, and it was calculated using only frames where flies moved above 4 mm/s (which corresponds to two body lengths).

## Appetitive long-term memory test

Appetitive long-term memory was tested as described in *Krashes and Waddell, 2011*. The two conditioning odours used were isoamyl acetate (Sigma-Aldrich #24900822 6 mL in 8 mL mineral oil; Sigma-Aldrich #330760) and 4-methylcyclohexanol (Sigma-Aldrich #153095, 10 mL in 8 mL mineral oil). Groups of 80–120 mixed sex flies were starved in a 1% agar tube filled with a damp 20 × 60 mm piece of filter paper for 18–20 hr before conditioning. During conditioning training, one odorant was presented with a dry filter paper (unconditioned odour, CS-) for 2 min, before a 30 s break, and presentation of a second odorant with filter paper coated with dry sucrose (conditioned odour, CS+). The test was repeated pairing the other odorant with sucrose, with a different group of flies to form one replicate. After training, flies were transferred back to agar tubes for testing 24 h later. PI was calculated in the same way as in *Krashes and Waddell, 2011*, as the number of flies approaching the conditioned odour minus the number of flies going in the opposite direction, divided by the total number of flies. A single PI value is the average score from the test with the reverse conditioning odour combination. Groups for which the total number of flies among both odorants was below 15 were discarded. For the *DNT-2* over-expression experiments with *tubGAL80$^{ts}$; DNT-2>DNT-2FL*, flies were raised at 18°C until 7–9 days post-eclosion. They were then either transferred to and maintained at 23°C (controls) or 30°C for 18–20 hr starvation, training, and up to testing 24 hr later.

## Statistical analysis

Statistical analyses were carried out using GraphPad Prism. CI was 95%, setting significance at p<0.05. Chi-square tests were carried out when comparing categorical data. Numerical data were tested first for their type of distributions. If data were distributed normally, unpaired Student's *t*-tests were used to compare means between two groups and one-way ANOVA or Welch ANOVA for larger groups, followed by post hoc Dunnett's test for multiple comparisons to a fixed control. Two-way ANOVA was used when comparisons to two variables were made. If data were not normally distributed, non-parametric Mann–Whitney *U*-test for two two group comparisons and Kruskal–Wallis ANOVA for larger groups, followed by post hoc Dunn's multiple comparisons test to a fixed control. Statistical details, including full genotypes, sample sizes, tests, and p-values, are provided in *Supplementary file 2*.

## Acknowledgements

We thank our lab, Carolina Rezaval, and Thomas Riemensperger for comments on the manuscript; Carolina Rezaval, Reinhard Wolf, Martin Heisenberg, and Scott Waddell for advice; Karina Piotrowska for help with behaviour experiments; Xiufeng Li for help with programming; Serge Birman, Ann-Shyn Chiang, Ron Davis, André Fialá, Barret Pfeiffer, Xi Rao, Carolina Rezaval, and Iris Salecker for flies; DSHB (Iowa) for antibodies; AddGene for plasmids; and Bloomington Drosophila Stock Center for *Drosophila* stocks. This work was funded by Marie-Curie Sklodowska Post-Doctoral fellowship to JS; Science Without Borders-CAPES PhD Studentship BEX 13380/13-3 to SUB; BBSRC Project Grants BB/R00871X/1 and BB/P004997/1 to AH; and Wellcome Trust Investigator Award 223197/Z/21/Z to AH.

## Additional information

### Funding

| Funder | Grant reference number | Author |
| --- | --- | --- |
| HORIZON EUROPE Marie Sklodowska-Curie Actions | TOLKEDA | Jun Sun |
| Coordenação de Aperfeiçoamento de Pessoal de Nível Superior | SWB PhD Scholarship | Suzana Ulian-Benitez |
| Wellcome Trust | 223197/Z/21/Z | Alicia Hidalgo |
| Darwin Trust of Edinburgh | PhD Studentship | Deepanshu ND Singh |
| Biotechnology and Biological Sciences Research Council | BB/R00871X/1 | Alicia Hidalgo |
| Biotechnology and Biological Sciences Research Council | BB/P004997/1 | Alicia Hidalgo |

The funders had no role in study design, data collection and interpretation, or the decision to submit the work for publication. For the purpose of Open Access, the authors have applied a CC BY public copyright license to any Author Accepted Manuscript version arising from this submission.

### Author contributions

Jun Sun, Conceptualization, Data curation, Formal analysis, Funding acquisition, Validation, Investigation, Visualization, Methodology, Writing - original draft, Writing – review and editing; Francisca Rojo-Cortes, Conceptualization, Data curation, Formal analysis, Validation, Investigation, Visualization, Methodology, Writing - original draft, Writing – review and editing; Suzana Ulian-Benitez, Data curation, Formal analysis, Investigation, Methodology, Writing – review and editing; Manuel G Forero, Software, Investigation, Methodology, Writing – review and editing; Guiyi Li, Data curation, Formal analysis, Validation, Investigation, Visualization, Methodology, Writing – review and editing; Deepanshu ND Singh, Data curation, Validation, Investigation, Visualization, Methodology, Writing – review and editing; Xiaocui Wang, Data curation, Formal analysis, Validation, Investigation, Methodology, Writing – review and editing; Sebastian Cachero, Resources, Investigation, Methodology, Writing – review and editing; Marta Moreira, Data curation, Investigation, Methodology, Writing – review and editing; Dean Kavanagh, Supervision, Methodology, Writing – review and editing; Gregory SXE Jefferis, Supervision, Writing – review and editing; Vincent Croset, Conceptualization, Supervision, Investigation, Writing - original draft, Writing – review and editing; Alicia Hidalgo, Conceptualization, Data curation, Formal analysis, Supervision, Funding acquisition, Validation, Investigation, Methodology, Writing - original draft, Project administration, Writing – review and editing

### Author ORCIDs

Francisca Rojo-Cortes ⓘ https://orcid.org/0000-0003-2332-8423
Manuel G Forero ⓘ https://orcid.org/0000-0001-9972-8621
Guiyi Li ⓘ https://orcid.org/0000-0001-9620-5139

Marta Moreira https://orcid.org/0000-0002-4779-4077
Gregory SXE Jefferis https://orcid.org/0000-0002-0587-9355
Vincent Croset https://orcid.org/0000-0001-9696-766X
Alicia Hidalgo https://orcid.org/0000-0001-8041-5764

Reviewer #1 (Public review): https://doi.org/10.7554/eLife.102222.3.sa1
Reviewer #2 (Public review): https://doi.org/10.7554/eLife.102222.3.sa2
Reviewer #3 (Public review): https://doi.org/10.7554/eLife.102222.3.sa3
Author response https://doi.org/10.7554/eLife.102222.3.sa4

## Additional files

### Supplementary files

• Supplementary file 1. Expression of *Tolls, keks,* and Toll downstream adaptors in cells related to DNT-2A neurons. Genes expressed in DNT-2 neurons, their potential and/or experimentally verified inputs and outputs, were identified with a combination of reporters (this work) and data from public single-cell RNAseq databases.

• Supplementary file 2. Statistical analysis. Table provides full genotypes, sample sizes, statistical tests, multiple comparison corrections, and p-values.

• MDAR checklist

### Data availability

All data generated or analysed during this study are included in the manuscript and supporting files; source data files have been provided for Figures 2, 3, 4, 5, 6 and their figure supplements.

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
