## [Editor Report · eLife Assessment]

This **important** study identifies neurotrophin signalling as a molecular mechanism underlying previous findings of structural plasticity in central dopaminergic neurons of the adult fly brain. The authors present **solid** evidence for neurotrophin signalling in shaping the structure and synapses of certain dopaminergic circuits. The work suggests an intriguing potential link between neurotrophin signaling and experience-induced structural plasticity, but further research will be necessary to establish this connection definitively.

---

## [Referee Report · Reviewer #1 (Public review)]

Summary:

Sun et al. are interested in how experience can shape the brain and specifically investigate the plasticity of the Toll-6 receptor-expressing dopaminergic neurons (DANs). To learn more about the role of Toll-6 in the DANs, the authors examine the expression of the Toll-6 receptor ligand, DNT-2. They show that DNT-2 expressing cells connect with DANs and that loss of function of DNT-2 in these cells reduces the number of PAM DANs, while overexpression causes alterations in dendrite complexity. Finally, the authors show that alterations in the levels of DNT-2 and Toll-6 can impact DAN-driven behaviors such as climbing, arena locomotion, and learning and long-term memory.

Strengths:

The authors methodically test which neurotransmitters are expressed by the 4 prominent DNT-2 expressing neurons and show that they are glutamatergic. They also use Trans-Tango and Bac-TRACE to examine the connectivity of the DNT-2 neurons to the dopaminergic circuit and show that DNT-2 neurons receive dopaminergic inputs and output to a variety of neurons including MB Kenyon cells, DAL neurons, and possibly DANS.

Weaknesses:

(1) To identify the DNT-2 neurons, the authors use CRISPR to generate a new DN2-GAL4. They note that they identified at least 12 DNT-2 plus neurons. In Supplementary Figure 1A, the DNT-2-GAL4 driver was used to express a UAS-histoneYFP nuclear marker. From these figures, it looks like DNT-2-GAL4 is labeling more than 12 neurons. Is there glial expression? This question is relevant as it is not clear how many other cell types are being manipulated with the DNT-2-GAL4 driver is used in subsequent experiments. For example, is DNT-2-GAL4 DNT-2-RNAi is reducing DNT2 in many neurons or glia effects could be indirect.

(2) In Figure 2C the authors show that DNT-2 upregulation leads to an increase in TH levels using q-RT-PCR from whole heads. However, in Figure 3G they also show that DNT-2 overexpression also causes an increase in the number of TH neurons. It is unclear whether TH RNA increases due to expression/cell or number of TH neurons in the head.

(3)DNT-2 is also known as Spz5 and has been shown to activate Toll-6 receptors in glia (McLaughlin et al., 2019), resulting in the phagocytosis of apoptotic neurons. In addition, the knock-down of DNT-2/Spz5 throughout development causes an increase in apoptotic debris in the brain, which can lead to neurodegeneration. Indeed Figure 3H shows that an adult-specific knock-down of DNT-2 using DNT2-GAL4 causes an increase in Dcp1 signal in many neurons and not just TH neurons.

Comments on revisions:

The authors have made some changes in the text to tone down their claims. They have also provided additional images to support their work. However, requested controls are not provided, and new experiments are not added to address reviewer concerns.

---

## [Referee Report · Reviewer #2 (Public review)]

This paper examines how structural plasticity in neural circuits, particularly in dopaminergic systems, is regulated by *Drosophila* neurotrophin-2 (DNT-2) and its receptors, Toll-6 and Kek-6. The authors show that these molecules are critical for modulating circuit structure, dopaminergic neuron survival, synaptogenesis, and connectivity. They demonstrate that the loss of DNT-2 or Toll-6 function leads to the loss of dopaminergic neurons, reduced dendritic arborization, and synaptic impairment, whereas overexpression of DNT-2 increases dendritic complexity and synaptogenesis. Additionally, DNT-2 and Toll-6 influence dopamine-dependent behaviors, including locomotion and long-term memory, suggesting a link between DNT-2 signaling, structural plasticity, and behavior.

A major strength of this study is the impressive cellular resolution achieved. By focusing on specific dopaminergic neurons, such as the PAM and PPL1 clusters, and using a range of molecular markers, the authors were able to clearly visualize intricate details of synapse formation, dendritic complexity, and axonal targeting within defined circuits. Given the critical role of dopaminergic pathways in learning and memory, this approach provides a valuable foundation for exploring the role of DNT-2, Toll-6, and Kek-6 in experience-dependent structural plasticity. While the manuscript hints at a connection to experience-induced plasticity, the study does not establish a direct causal link between neurotrophin signaling and experience-driven changes. To support this idea, it would be necessary to observe experience-induced structural changes and demonstrate that downregulation of DNT-2 signaling prevents these changes. The closest attempt in this study was the artificial activation of DNT-2 neurons using TrpA1, which resulted in overgrowth of axonal arbors and an increase in synaptic sites in both DNT-2 and PAM neurons. However, whether the observed structural changes were dependent on DNT-2 signaling remains unclear.

In conclusion, this study demonstrates that DNT-2 and its receptors play a role in regulating the structure of dopaminergic circuits in the adult fly brain. Whether DNT-2 signaling contributes to experience-dependent structural plasticity within these circuits remains an exciting open question and warrants further investigation.

Comments on revisions:

I appreciate the authors' responses to my previous comments and have no further suggestions.

---

## [Referee Report · Reviewer #3 (Public review)]

Summary:

The authors used the model organism *Drosophila melanogaster* to show that the neurotrophin Toll-6 and its ligands, DNT-2 and kek-6, play a role in maintaining the number of dopaminergic neurons and modulating their synaptic connectivity. This supports previous findings on the structural plasticity of dopaminergic neurons and suggests a molecular mechanism underlying this plasticity.

Strengths:

The experiments are overall very well designed and conclusive. Methods are in general state-of-the-art, the sample sizes are sufficient, the statistical analyses are sound, and all necessary controls are at place. The data interpretation is straight forwards, and the relevant literature is taken into consideration. Overall, the manuscript is solid and presents novel, interesting and important findings.

Weaknesses:

There are three technical weaknesses that could perhaps be improved.

First, the model of reciprocal, inhibitory feedback loops (figure 2F) is speculative. On the one hand, glutamate can act in flies as excitatory or inhibitory transmitter (line 157!), and either situation can be the case here. On the other hand, it is not clear how an increase or decrease in cAMP level translates into transmitter release. One can only conclude that two type of neurons potentially influence each other.

Second, the quantification of bouton volumes (no y-axis label in Figure 5 C and D!) and dendrite complexity are not convincingly laid out. Here, the reader expects fine-grained anatomical characterizations of the structures under investigation, and a method to precisely quantify the lengths and branching patterns of individual dendritic arborizations as well as the volume of individual axonal boutons.

Third, figure 1C shows two neurons with the goal of demonstrating between-neuron variability. It is not convincingly demonstrated that the two neurons are actually of the very same type of neuron in different flies, or two completely different neurons.

Review of the revised manuscript:

The authors have addressed some points of concern raised by the reviewers. I would like to emphasize that I find the overall research study highly interesting and important.

---

## [Author Response]

The following is the authors’ response to the original reviews.

**Reviewer #1 (Public review):**
Summary:Sun et al. are interested in how experience can shape the brain and specifically investigate the plasticity of the Toll-6 receptor-expressing dopaminergic neurons (DANs). To learn more about the role of Toll-6 in the DANs, the authors examine the expression of the Toll-6 receptor ligand, DNT-2. They show that DNT-2 expressing cells connect with DANs and that loss of function of DNT-2 in these cells reduces the number of PAM DANs, while overexpression causes alterations in dendrite complexity. Finally, the authors show that alterations in the levels of DNT-2 and Toll-6 can impact DAN-driven behaviors such as climbing, arena locomotion, and learning and long-term memory.Strengths:The authors methodically test which neurotransmitters are expressed by the 4 prominent DNT-2 expressing neurons and show that they are glutamatergic. They also use Trans-Tango and Bac-TRACE to examine the connectivity of the DNT-2 neurons to the dopaminergic circuit and show that DNT-2 neurons receive dopaminergic inputs and output to a variety of neurons including MB Kenyon cells, DAL neurons, and possibly DANS.

We are very pleased that Reviewer 1 found our connectivity analysis a strength.

Weaknesses:(1) To identify the DNT-2 neurons, the authors use CRISPR to generate a new DN2-GAL4.They note that they identified at least 12 DNT-2 plus neurons. In Supplementary Figure 1A, the DNT-2-GAL4 driver was used to express a UAS-histoneYFP nuclear marker. From these figures, it looks like DNT-2-GAL4 is labeling more than 12 neurons. Is there glial expression?

Indeed, we claimed that DNT-2 is expressed in at least 12 neurons (see line 141, page 6 of original manuscript), which means more than 12 could be found. The membrane tethered reporters we used – UAS-FlyBow1.1, UASmcD8-RFP, UAS-MCFO, as well as UAS-DenMark:UASsyd-1GFP – gave a consistent and reproducible pattern. However, with DNT-2GAL4>UAS-Histone-YFP more nuclei were detected that were not revealed by the other reporters. We have found also with other GAL4 lines that the patterns produced by different reporters can vary. This could be due to the signal strength (eg His-YFP is very strong) and perdurance of the reporter (e.g. the turnover of His-YFP may be slower than that of the other fusion proteins).

We did not test for glial expression, as it was not directly related to the question addressed in this work.

(2) In Figure 2C the authors show that DNT-2 upregulation leads to an increase in TH levels using q-RT-PCR from whole heads. However, in Figure 3H they also show that DNT-2 overexpression also causes an increase in the number of TH neurons. It is unclear whether TH RNA increases due to expression/cell or the number of TH neurons in the head.

Figure 3H shows that over-expression of DNT-2 FL increased the number of Dcp1+ apoptotic cells in the brain, but not significantly (p=0.0939). The ability of full-length neurotrophins to induce apoptosis and cleaved neurotrophins promote cell survival is well documented in mammals. We had previously shown that DNT-2 is naturally cleaved, and that over-expression of DNT-2 does not induce apoptosis in the various contexts tested before (McIlroy et al 2013 Nature Neuroscience; Foldi et al 2017 J Cell Biol; Ulian-Benitez et al 2017 PLoS Genetics). Similarly, throughout this work we did not find DNT-2FL to induce apoptosis.

Instead, in Figure 3G we show that over-expression of DNT-2FL causes a statistically significant increase in the number of TH+ cells. This is an important finding that supports the plastic regulation of PAM cell number. We thank the Reviewer for highlighting this point, as we had forgotten to add the significance star in the graph. In this context, we cannot rule out the possibility that the increase in TH mRNA observed when we over-express DNT-2FL could not be due to an increase in cell number instead. Unfortunately, it is not possible for us to separate these two processes at this time. Either way, the result would still be the same: an increase in dopamine production when DNT-2 levels rise.

We have now edited the abstract lines 38-39 adding that “By contrast, over-expressed DNT-2 increased DAN cell number,…”, within the main text in Results page 10 lines 259-265 and in the Discussion section page 15 lines 391, 393-396.

(3) DNT-2 is also known as Spz5 and has been shown to activate Toll-6 receptors in glia (McLaughlin et al., 2019), resulting in the phagocytosis of apoptotic neurons. In addition, the knock-down of DNT-2/Spz5 throughout development causes an increase in apoptotic debris in the brain, which can lead to neurodegeneration. Indeed Figure 3H shows that an adult specific knock-down of DNT-2 using DNT2-GAL4 causes an increase in Dcp1 signal in many neurons and not just TH neurons.

Indeed, we did find Dcp1+ TH-negative cells too (although not widely throughout the brain), although this is not shown in the images of Figure 3H where we showed only TH+ Dcp+ cells.

That is not surprising, as DNT-2 neurons have large arborisations that can reach a wide range of targets; DNT-2 is secreted, and could reach beyond its immediate targets; Toll-6 is expressed in a vast number of cells in the brain; DNT-2 can bind promiscuously at least also Toll-7 and other Keks, which are also expressed in the adult brain (Foldi et al 2017 J Cell Biology; Ulian-Benitez et al 2017 PLoS Genetics; Li et al 2020 eLife). Together with the findings by McLaughlin et al 2019, our findings further support the notion that DNT-2 is a neuroprotective factor in the adult brain. It will be interesting to find out what other neuron types DNT-2 maintains.

We have made some edits on these points in page 10 lines 259-265.

We would like to thank Reviewer 1 for their positive comments on our work and their interesting and valuable feedback.

**Reviewer #2 (Public review):**
This paper examines how structural plasticity in neural circuits, particularly in dopaminergic systems, is regulated by *Drosophila* neurotrophin-2 (DNT-2) and its receptors, Toll-6 and Kek-6. The authors show that these molecules are critical for modulating circuit structure and dopaminergic neuron survival, synaptogenesis, and connectivity. They show that loss of DNT-2 or Toll-6 function leads to loss of dopaminergic neurons, dendritic arborization, and synaptic impairment, whereas overexpression of DNT-2 increases dendritic complexity and synaptogenesis. In addition, DNT-2 and Toll-6 modulate dopamine-dependent behaviors, including locomotion and long-term memory, suggesting a link between DNT-2 signaling, structural plasticity, and behavior.A major strength of this study is the impressive cellular resolution achieved. By focusing on specific dopaminergic neurons, such as the PAM and PPL1 clusters, and using a range of molecular markers, the authors were able to clearly visualize intricate details of synapse formation, dendritic complexity, and axonal targeting within defined circuits. Given the critical role of dopaminergic pathways in learning and memory, this approach provides a good opportunity to explore the role of DNT-2, Toll-6, and Kek-6 in experience-dependent structural plasticity. However, despite the promise in the abstract and introduction of the paper, the study falls short of establishing a direct causal link between neurotrophin signaling and experience-induced plasticity.Simply put, this study does not provide strong evidence that experience-induced structural plasticity requires DNT-2 signaling. To support this idea, it would be necessary to observe experience-induced structural changes and demonstrate that downregulation of DNT-2 signaling prevents these changes. The closest attempt to address this in this study was the artificial activation of DNT-2 neurons using TrpA1, which resulted in overgrowth of axonal arbors and an increase in synaptic sites in both DNT-2 and PAM neurons. However, this activation method is quite artificial, and the authors did not test whether the observed structural changes were dependent on DNT-2 signaling. Although they also showed that overexpression of DNT-2FL in DNT-2 neurons promotes synaptogenesis, this phenotype was not fully consistent with the TrpA1 activation results (Figures 5C and D).In conclusion, this study demonstrates that DNT-2 and its receptors play a role in regulating the structure of dopaminergic circuits in the adult fly brain. However, it does not provide convincing evidence for a causal link between DNT-2 signaling and experience-dependent structural plasticity within these circuits.

We would like to thank Reviewer 2 for their very positive assessment of our approach to investigate structural circuit plasticity. We are delighted that this Reviewer found our cellular resolution impressive. We are also very pleased that Reviewer 2 found that our work demonstrates that DNT-2 and its receptors regulate the structure of dopaminergic circuits in the adult fly brain. This is already a very important finding that contributes to demonstrating that, rather than being hardwired, the adult fly brain is plastic, like the mammalian brain. Furthermore, it is remarkable that this involves a neurotrophin functioning via Toll and kinase-less Trks, opening an opportunity to explore whether such a mechanism could also operate in the human brain.

We are very pleased that this Reviewer acknowledges that this work provides a good opportunity to explore the role of DNT-2, Toll-6, and Kek-6 in experience-dependent structural plasticity. We provide a molecular mechanism and proof of principle, and we demonstrate a direct link between the function of DNT-2 and its receptors in circuit plasticity. We also showed a link of DNT-2 to neuronal activity, as neuronal activity increased the production of DNT-2GFP, induced the cleavage of DNT-2 and a feedback loop between DNT-2 and dopamine, and both neuronal activity and increased DNT-2 levels promoted synaptogenesis.

As the Reviewer acknowledges this approach provides a good opportunity to explore the role of DNT-2, Toll-6, and Kek-6 in experience-dependent structural plasticity. Finding out the direct link in response to lived experience is a big task, beyond the scope of this manuscript, and we will be testing this with future projects. Nevertheless, it is important to place our findings within this context together with the link to mammalian neurotrophins (as explained in the discussion), as it is here where the findings have deep and impactful implications.

To accommodate the criticism of this Reviewer, we have now toned down our narrative. This does not diminish the importance of the findings, it makes the argument more stringent. Please see edits in: Abstract page 2 lines 42-44; and Discussion page 22 line 586 – which were the only points were a direct claim had been made.

We would like to thank Reviewer 2 for the positive and thoughtful evaluation of our work, and for their feedback.

**Reviewer #3 (Public review):**
Summary:The authors used the model organism *Drosophila melanogaster* to show that the neurotrophin Toll-6 and its ligands, DNT-2 and kek-6, play a role in maintaining the number of dopaminergic neurons and modulating their synaptic connectivity. This supports previous findings on the structural plasticity of dopaminergic neurons and suggests a molecular mechanism underlying this plasticity.Strengths:The experiments are overall very well designed and conclusive. Methods are in general state-of-the-art, the sample sizes are sufficient, the statistical analyses are sound, and all necessary controls are in place. The data interpretation is straightforward, and the relevant literature is taken into consideration. Overall, the manuscript is solid and presents novel, interesting, and important findings.

We are delighted that Reviewer 3 found our work solid, novel, interesting and with important findings. We are also very pleased that this Reviewer found that all necessary controls have been carried out.

Weaknesses:There are three technical weaknesses that could perhaps be improved.First, the model of reciprocal, inhibitory feedback loops (Figure 2F) is speculative. On the one hand, glutamate can act in flies as an excitatory or inhibitory transmitter (line 157), and either situation can be the case here. On the other hand, it is not clear how an increase or decrease in cAMP level translates into transmitter release. One can only conclude that two types of neurons potentially influence each other.

Thank you for pointing out that glutamate can be inhibitory. In response, we have removed the word ‘excitatory’ from the only point it had been used in the text: page 7 line 167.

In mammals, the neurotrophin BDNF has an important function in glutamatergic synapses, thus we were intrigued by a potential evolutionary conservation. Our evidence that DNT-2A neurons could be excitatory is indirect, yet supportive: exciting DNT-2 neurons with optogenetics resulted in an increase in GCaMP in PAMs (data not shown); over-expression of DNT-2 in DNT-2 neurons increased TH mRNA levels; optogenetic activation of DNT-2 neurons results in the Dop2R-dependent downregulation of cAMP levels in DNT-2 neurons. Dop2R signals in response to dopamine, which would be released only if dopaminergic neurons had been excited. Accordingly, glutamate released from DNT-2 neurons would have been rather unlikely to inhibit DANs.

cAMP is a second messenger that enables the activation of PKA. PKA phosphorylates many target proteins, amongst which are various channels. This includes the voltage gated calcium channels located at the synapse, whose phosphorylation increases their opening probability. Other targets regulate synaptic vesicle release. Thus, a rise in cAMP could facilitate neurotransmitter release, and a downregulation would have the opposite effect. Other targets of PKA include CREB, leading to changes in gene expression. Conceivably, a decrease in PKA activity could result in the downregulation of DNT-2 expression in DNT-2 neurons. This negative feedback loop would restore the homeostatic relationship between DNT-2 and dopamine levels.

We agree with this Reviewer that whereas our qRT-PCR data show that over-expression of DNT-2 increases TH mRNA levels, this does not demonstrate that originates from PAM neurons. Similarly, although our EPAC data imply that dopamine must be released from DANs and received by DNT-2 neurons to explain those data, the evidence did not include direct visualisation of dopamine release in response to DNT-2 neuron activation. To accommodate these criticisms, we have edited the summary Figure 2E adding question marks to indicate inference points and page 9 line 221.

Our data indeed demonstrate that DNT-2 and PAM neurons influence each other, not potentially, but really. We have provided data that: DNT-2 and PAMs are connected through circuitry; that the DNT-2 receptors Toll-6 and kek-6 are expressed in DANs, including in PAMs; that alterations in the levels of DNT-2 (both loss and gain of function) and loss of function for the DNT-2 receptors Toll-6 and Kek-6 alter PAM cell number, alter PAM dendritic complexity and alter synaptogenesis in PAMs; alterations in the levels of DNT-2, Toll-6 and kek-6 in adult flies alters dopamine dependent behaviours of climbing, locomotion in an arena and learning and long-term memory. These data firmly demonstrate that the two neuron types DNT-2 and PAMs influence each other.

We have also shown that over-expression of DNT-2 in DNT-2 neurons increases TH mRNA levels, whereas activation of DNT-2 neurons decreases cAMP levels in DNT-2 neurons in a dopamine/Dop2R-dependent manner. These data show a functional interaction between DNT-2 and PAM neurons.

Second, the quantification of bouton volumes (no y-axis label in Figure 5 C and D!) and dendrite complexity are not convincingly laid out. Here, the reader expects fine-grained anatomical characterizations of the structures under investigation, and a method to precisely quantify the lengths and branching patterns of individual dendritic arborizations as well as the volume of individual axonal boutons.

Figure 5C, D do contain Y-axis labels, all our graphs in main manuscript and in supplementary files contain Y-axis labels.

In fact, we did use a method to precisely quantify the lengths and branching patterns of individual dendritic arborisations, volume of individual boutons and bouton counting. These analyses were carried out using Imaris software. For dendritic branching patterns, the “Filament Autodetect” function was used. Here, dendrites were analysed by tracing semi-automatically each dendrite branch (ie manual correction of segmentation errors) to reconstruct the segmented dendrite in volume. From this segmented dendrite, Imaris provides measurements of total dendrite volume, number and length of dendrite branches, terminal points, etc. For bouton size and number, we used the Imaris “Spot” function. Here, a threshold is set to exclude small dots (eg of background) that do not correspond to synapses/boutons. All samples and genotypes are treated with the same threshold, thus the analysis is objective and large sample sizes can be analysed effectively. We had already provided a description of the use of Imaris in the methods section.

We have now exapanded the protocol on how we use Imaris to analyse dendrites and synapses, in: Materials and Methods section, page 28 lines 756-768 and page 29 lines 778-799.

Third, Figure 1C shows two neurons with the goal of demonstrating between-neuron variability. It is not convincingly demonstrated that the two neurons are actually of the very same type of neuron in different flies or two completely different neurons.

We thank Reviewer 3 for raising this interesting point. It is not possible to prove which of the four DNT-2A neurons per hemibrain, which we visualised with DNT-2>MCFO, were the same neurons in every individual brain we looked at. This is because in every brain we have looked at, the soma of the neurons were not located in exactly the same location. Furthermore, the arborisation patterns are also different and unique, for each individual brain. Thus, there is natural variability in the position of the soma and in the arborisation patterns. Such variability presumably results from the combination of developmental and activity-dependent plasticity. Importantly, for every staining we carried out using DNT-2GAL4 and various membrane reporters and MCFO clones, we never found two identical DNT-2 neuron profiles.

To increase the evidence in support of this point, we have now expanded Figure 1, adding one more image of DNT-2>FlyBow (Figure 1A) and two more images of DNT-2>MCFO (Figure 1D). In total, seven images in Figure 1 and two further images in Figure 5A demonstrate the variability of DNT-2 neurons.

We would like to thank Reviewer 3 for the very positive evaluation of our work and the interesting and valuable feedback.

**Recommendations for the authors:**

**Reviewer #1 (Recommendations for the authors):**
In the fly list, several fly lines are missing references and sources.

Apologies for this over-sight, this has now been corrected.

We thank Reviewer 1 for their effort and time to scrutinise our work, and for their very positive and helpful feedback.

**Reviewer #2 (Recommendations for the authors):**
(1) Here I provide some more specific comments that I hope will help the authors further improve the study.(2) L148: "single neuron clones revealed variability in the DNT-2A". How do the authors know that they are labeling the same subtype of DNT-2A neurons?

There are four anterior DNT-2A cells per hemibrain, that project from the SOG area to the SMP. It is not possible to verify that every time we look at exactly the same neuron, because the exact position of the somas and the arborisation patterns vary from brain to brain. We know this from two sources of data: (1) when using DNT-2GAL4 to visualise the expression of membrane reporters (e.g. UAS-FlyBow, UAS-mCD8-GFP, UAS-CD8-RFP) no brain ever showed a pattern identical to that of another brain, neither in the exact position of the somas nor in the exact arborisation patterns. (2) When we generated DNT-2>MCFO clones to visualise 1-2 cells at a time, no single neuron or 2-neuron clones ever showed an identical pattern. The most parsimonious interpretation is that the exact location of the somas and the exact arborisation patterns vary across individual flies. Developmental variability in neuronal patterns has also been reporter by Linneweber et al (2020) Science.

To make our evidence more compelling, and in response to this Reviewer’s query, we have now added further images. Please find in revised Figure 1 A,B three examples of three different brains expressing DNT-2>FlyBow1.1. In Figure 1D, two more examples (altogether 4) of DNT-2>MCFO clones. Here it is clear to see that no neuron shape is identical to that of others, demonstrating variability in individual fly brains. We now show four images in Figure 1 and two more in Figure 5A that demonstrate the variability of DNT-2A neurons.

(3) Figure 1E: Are all DNT-2A neurons positive for vGlut and Dop2R? This figure shows only two DNT-2A neurons.

Yes, all four DNT-2A neurons per hemibrain are vGlut positive and we have now added more images to Supplementary Figure S1A (right), also showing that presynaptic DNT-2A endings at SMP also coincide with a vGlut+ domain (Figure S1A left).

Yes, all all four DNT-2A neurons per hemibrain are Dop2R positive and we have now added more images to Supplementary Figure S1B.

(4) L156: Glutamate is generally considered to be inhibitory in the adult fly brain. More evidence is needed before the authors can claim that "DNT-2A neurons are excitatory glutamatergic neurons".

Thank you for pointing this out. Although our data do not conclusively demonstrate it, they are consistent with DNT-2A neurons being excitatory. BDNF is most commonly released from glutamatergic neurons in mammals, its release is activity-dependent and leads to formation and stabilisation of synapses. The phenotypes we have observed are consistent with this and reveal functional evolutionarily conservation: (1) exciting DNT-2 neurons with TrpA1 results in increased production and cleavage of DNT-2GFP and de novo synaptogenesis; (2) over-expression of DNT-2 in the adult induces de novo synaptogenesis; (3) down-regulation or loss of DNT-2 and its receptors Toll-6 and Kek-6 impair synaptogenesis. Furthermore, we show that DNT-2 dependent synaptogenesis is between DNT-2 and dopaminergic neurons, which are involved in the control of locomotion, reward learning and long-term memory, and dopamine itself is required for such behaviour. Consistently with this we found that: (1) over-expression of DNT-2 increases TH mRNA levels, which would lead to the up-regulation of dopamine production; (2) exciting DNT-2 neurons increases locomotion speed in an arena; (3) knock-down of DNT-2 and its receptors decreases locomotion, whereas over-expression of DNT-2 increases locomotion; (4) over-expression of DNT-2 increases learning and long-term memory. Finally, in a previous version in bioRxiv, we also showed using optogenetics and calcium imaging that exciting DNT-2 neurons induced GCaMP signalling in their output PAM neurons, and in this version we show that exciting DNT-2 neurons regulates cAMP in DNT-2 neurons via dopamine-release dependent feedback. Altogether, the most parsimonious interpretation of these data is that vGlut+ DNT-2 neurons are excitatory.

In any case, to address this reviewer’s point, we have now removed the word ‘excitatory’ from page 7 line 167.

(5) Figure 1H, I: A more detailed description of the Toll-6 and Kek-6 expressing neurons will be helpful. Are they expressed in specific types of PAM and PPL1 DANs? The legend in Figure S2 mentions labeling in γ2α′1 zones, but it seems to be more than that.

This information had been already provided, presumable this Reviewer overlooked this. This was already described in great detail by comparing our microscopy data with the single cell RNA-seq data available through Fly Cell Atlas (https://flycellatlas.org) and Scope (https://scope.aertslab.org/#/b77838f4-af3c-4c37-8dd9-cf7a41e4b034/*/welcome).

Please see our previously submitted Table S1 “Expression of Tolls, keks and Toll downstream adaptors in cells related to DNT-2A neurons”.

(6) Figure S3 should be controls for Figure 2A. It is incorrectly labeled as controls for Figure 3A.

Thank you for pointing out this typo, this has now been corrected.

(7) L197: The authors state, "This showed that DNT-2 could stimulate dopamine production in neighboring DANs". However, the results do not fully support this conclusion because the experiments measure overall TH levels in the brain, not specifically in neighboring DANs. The observed effect could be indirect via other neurons.

Indeed, we have now edited the text to: “This showed that DNT-2 could stimulate dopamine production”: page 8 line 208.

(8) Figure 3: If Toll-6 is expressed in specific subtypes of PAM DANs, are they the dying cells when Toll-6 was knocked down? I think the paper will be significantly improved if the authors provide a more in-depth analysis of the phenotype. Also, permissive temperature controls are missing for the experiments in (E)-(H). Permissive controls are essential to confirm that the observed effects are due to adult-specific RNAi knock-down.

Current tools do not enable us to visualise Toll-6+ neurons at the same time as manipulating DNT-2 neurons and at the same time as monitoring Dcp1. Stainings with Dcp1 in the adult brain are not trivial. Thus, we cannot guarantee this. However, Toll-6 is the preferential receptor for DNT-2, and given that apoptosis increases when we knock-down DNT-2, the most parsimonious interpretation is that the dying cells bear the DNT-2 receptor Toll-6. Even if DNT-2 can promiscuously bind other Toll receptors, the simplest way to interpret these data remains that DNT-2 promotes cell survival by signalling via its receptors, as no other possible route is known to date. This would be consistent with all other data in this figure.

We thank this Reviewer for the feedback on the controls. Unfortunately, these are not trivial experiments, they require considerable time, effort, dedication and skill. This manuscript has already taken 5 years of daily hard work. We no longer have the staff (ie the first author left the lab) nor resources to dedicate to address this point.

(9) Figure 4B: This phenotype in DNT-2 mutants is very striking. Did the neurons still survive and did their axonal innervation in the lobes remain intact?

Homozygous DNT-2 mutants are viable and have impair climbing, as we had already shown in Figure 7C.

(10) L261: The authors mention that "PAM-β2β′2 neurons express Toll-6 (Table S1)". However, I cannot find this information in Table S1.

Unfortunately, I cannot identify the source of that statement at present and the first authors has left the lab. In any case, although the fact that knocking down Toll-6 in these neurons causes a phenotype means they must, it does not directly prove it. We have now corrected this to: “PAM-b2b'2 neuron dendrites overlap axonal DNT2 projections”, page 11 line 280.

(11) Figure 4C, D: What about their synaptogenesis? Do they agree with the result in Figure 4B?

This was not tested at the time. Unfortunately, these are not trivial experiments and require considerable time, effort, dedication and skill. Addressing this point experimentally is not possible for us at this point. In any case, given the evidence we already provide, it is highly unlikely they would alter the interpretation of our findings and the value of the discoveries already provided.

(12) L270: The authors state: "To ask whether DNT-2 might affect axonal terminals, we tested PPL1 axons." However, it is unclear why the focus was shifted to PPL1 neurons when similar analyses could have been performed on PAM DANs for consistency. In addition, it would be beneficial to assess dendritic arbor complexity and synaptogenesis in PPL1-γ1-pedc neurons to provide a more comprehensive comparison between PPL1 and PAM DANs. Performing parallel analyses on both neuron types would strengthen the study by providing insight into the generality and specificity of DNT-2 in different dopaminergic circuits.

The question we addressed with Figure 4 was whether the DNT-2 and its receptors could modify axons, dendrites and synapses, ie all features of neuronal plasticity. The reason we used PPL1-g1-pedc to analyse axonal terminals was because of their morphology, which offered a clearer opportunity to visualise axonal endings than PAMs did. An exhaustive analysis of PPL1-g1-pedc is beyond the scope of this work and not the central focus.

(13) Figure 4G lacks a permissive temperature control, which is essential to confirm that the observed effects are due to adult-specific RNAi knock-down.

We thank this Reviewer for this feedback, which we will bear in mind for future projects.

(14) Figure 5A requires quantification and statistical comparison.

We thank this Reviewer for this feedback. We did consider this, but the data are too variable to quantify and we decided it was best to present it simply as an observation, interesting nonetheless. This is consistent as well with the data in Figure 1, which we have now expanded with this revision, which show the natural variability in DNT-2 neurons.

(15) Figure 5B: Many green signals in the control image are not labeled as PSDs, raising concerns about the accuracy of the image analysis methods used for synapse identification. While I trust that the authors have validated their analysis approach, it would strengthen the study if they provided a clearer description or evidence of the validation process.

This was done using the Imaris “Spot function”, in volume. A threshold is set to exclude spots due to GFP background and select only synaptic spots. The selection of spots and quantification are done automatically by Imaris. All spots below the threshold are excluded, regardless of genotype and experimental conditions, rendering the analysis objective. We have now provided a detailed description of the protocol in the Materials and Methods section: page 29 lines 778-799.

(16) Figure 5C lacks genotype controls (i.e., DNT2-GAL4-only and UAS-TrpA1-only). These controls are essential because elevated temperatures alone, without activation of DNT2 neurons, could potentially increase Syt-GCaMP production, leading to an increase in the number of Syt+ synapses. Including these controls would help ensure that the observed effects are truly due to the activation of DNT2 neurons and not temperature-related artifacts.

We thank this Reviewer for this feedback, which we will bear in mind for future projects.

(17) L314-316: The authors state, "Here, the coincidence of... revealed that newly formed synapses were stable." I think this statement needs to be toned down because there is no evidence that these pre- and post-synaptic sites are functionally connected.

The Reviewer is correct that our data did not visualise together, in the same preparation and specimen, both pre- and post-synaptic sites. Still, given that PAMs have already been proved by others to be required for locomotion, learning and long-term memory, our data strongly suggest that synapses between them at the SMP are functionally connected.

Nevertheless, as we do not provide direct cellular evidence, we have now edited the text to tone down this claim: “Here, the coincidence of increased pre-synaptic Syt-GFP from PAMs and post-synaptic Homer-GFP from DNT-2 neurons at SMP suggests that newly formed synapses could be stable”, page 13 line 351.

(18) Figure 5D lacks permissive temperature controls. Also, the DNT-2FL overexpression phenotypes are different from the TpA1 activation phenotypes. The authors may want to discuss this discrepancy.

Regarding the controls, these are not appropriate for this data set. These data were all taken at a constant temperature of 25°C, there were no shifts, and therefore do not require a permissive temperature control. We thank this Reviewer for drawing our attention to the fact that we made a mistake drawing the diagram, which we have now corrected in Figure 5D.

Regarding the discrepancy, this had already been discussed in the Discussion section of the previously submitted version, page 19 Line 509-526. Presumably this Reviewer missed this before.

(19) Figure 6A, B lack permissive temperature controls. These controls are important if the authors want to claim that the behavioral defects are due to adult-specific manipulations. In addition, there is no statistical difference between the PAM-GAL4 control and the RNAi knock-down group. The authors should be careful when stating that climbing was reduced in the RNAi knock-down flies (L341-342).

We thank this Reviewer for this feedback, which we will bear in mind for future projects.

Point taken, but climbing of the tubGAL80ts, PAM>Toll-6RNAi flies was significantly different from that of the UAS-Toll-6RNAi/+ control.

(20) Figure 6C: It seems that the DAN-GAL4 only control (the second group) also rescued the climbing defect. The authors may want to clarify this point.

The phenotype for this genotype was very variable, but certainly very distinct from that of flies over-expressing Toll-6[CY].

We thank Reviewer 2 for their very thorough analysis of our paper that has helped improve the work.

**Reviewer #3 (Recommendations for the authors):**
Overall, the manuscript reports highly interesting and mostly very convincing experiments.

We are very grateful to this Reviewer for their very positive evaluation of our work.

Based on my comments under the heading "public review", I would like to suggest three possible improvements.First, the quantification of structural plasticity at the sub-cellular level should be explained in more detail and potentially improved. For example, 3D reconstructions of individual neurons and quantification of the structure of boutons and dendrites could be undertaken. At present, it is not clear how bouton volumes are actually recorded accurately.

Thank you for the feedback. The analyses of dendrites and synapses were carried out in 3D-volumes using Imaris “Filament” module and “Spot function”, respectively. Dendrites are analysed semi-automatically, ie correcting potential branching errors of Imaris, and synapses are counted automatically, after setting appropriate thresholds. Details have now been expanded in the Materials and Sections section: page 28 lines 756-768 and page 29 lines 780-799.

We would also like to thank Imaris for enabling and facilitating our remote working using their software during the Covid-19 pandemic, post-pandemic lockdowns and lab restrictions that spanned for over a year.

Second, the variability between DNT-2A-positive neurons with increasing sample size compared to a control (DNT-2A-negative neurons) should be demonstrated. Figure 2C does currently not present convincing evidence of increased structural variability.

It is unclear what data the Reviewer refers to. Figure 2C shows qRT-PCR data, and it does not show structural variability, which instead is shown with microscopy. If it is the BacTrace data in Figure 2B, the controls had been provided and the data were unambiguous. If Reviewer means Figure 1C, it is unclear why DNT-2GAL4-negative flies are needed when the aim was to visualise normal (not genetically manipulated) DNT-2 neurons. Thus, unfortunately we do not understand what the point is here.

The observation that DNT-2 neurons are very variable, naturally, is highly interesting, and presumably this is what drew the attention of Reviewer 3. We agree that showing further data in support of this is interesting and valuable. Thus, in response to this Reviewer’s comment we have now increased the number of images that demonstrate variability of DNT-2 neurons:

(1) We have added an extra image, altogether providing three images in new Figure 1A showing three different individual brains stained with DNT-2GAL4>UAS-FlyBow1.1. These show common morphology and features, but different location of the somas and distinct detailed arborisation patterns. Two more images using DNT-2GAL4 are provided in Figure 5A.

(2) We have now added two further MCFO images, altogether showing four examples where the somas are not always in the same location and the axons arborise consistently at the SMP, but the detailed projections are not identical: new Figure 1D.

These data compellingly show natural variability in DNT-2 neuron morphology.

Third, I propose to simplify the feedback model (Figure 2F) to be less speculative.

Indeed, some details in Figure 2F are speculative as we did not measure real dopamine levels. Accordingly, we have now edited this diagram, adding question marks to indicate speculative inference, to distinguish from the arrows that are grounded on the data we provide.

Accordingly, we have also edited the text in:

- page 9, lines 221: “Altogether, this shows that DNT-2 up-regulated TH levels (Figure 2E), and presumably via dopamine release, this inhibited cAMP in DNT-2A neurons (Figure 2F)”.

- page 20, lines 515: “Importantly, we showed that activating DNT-2 neurons increased the levels and cleavage of DNT-2, up-regulated DNT-2 increased TH expression, and this initial amplification resulted in the inhibition of cAMP signalling via the dopamine receptor Dop2R in DNT-2 neurons.”

As minor points:

(1) Appetitive olfactory learning is based on Tempel et al., (1983); Proc Natl Acad Sci U S A. 1983 Mar;80(5):1482-6. doi: 10.1073/pnas.80.5.1482. This paper should perhaps be cited.

Thank you for bringing this to our attention, we have now added this reference to page 14 line 394.

(2) Line 34: I would add ..."ligand for Toll-6 AND KEK-6,".

Indeed, thank you, now corrected.

(3) Line 39: DNT-2-POSITIVE NEURONS.

Now corrected, thank you.

(4) The levels of TH mRNA were quantified. Why not TH or dopamine directly using antibodies, ELISA, or HPLC? After all, later it is explicitly written that DNT modulates dopamine levels (line 481)!

We thank this Reviewer for this suggestion. We did try with HPLC once, but the results were inconclusive and optimising this would have required unaffordable effort by us and our collaborators. Part of this work spanned over the pandemic and subsequent lockdowns and lab restrictions to 30% then 50% lab capacity that continued for one year, making experimental work extremely challenging. Although we were unable to carry out all the ideal experiments, the DNT-2-dependent increase in TH mRNA coupled with the EPAC-Dop2R data provided solid evidence of a DNT-2-dopamine link.

(5) Line 271: The PPL1-g1-pedc neuron has mainly (but not excusively) a function in short-term memory!

They do, but others have also shown that PPL1-g1-pedc neurons have a gating function in long-term memory (Placais et al 2012; Placais et al 2017; Huang et al 2024) and are required for long-term memory (Adel and Griffith 2020; Boto et al 2020).

(6) Line 401: Reward learning requires PAM neurons. PPL1 neurons are required for aversive learning.

Indeed, PPL1 neurons are required for aversive learning, but they also have a gating function in long-term memory common for both reward and aversive learning (Adel and Griffith, 2020 Neurosci Bull; Placais et al, 2012 Nature Neuroscience; Placais et al 2017 Nature Communications; Huang et al 2024 Nature).

Overall, the manuscript presents extremely interesting, novel results, and I congratulate the authors on their findings.

We would like to thank this Reviewer for taking the time to scrutinise our work, their helpful feedback that has helped us improve the work and for their interest and positive and kind works.